# Modelling vegetation land fragmentation in urban areas of Western Province, Sri Lanka using an Artificial Intelligence-based simulation technique

Amila Jayasinghe⊙*, Nesha Ranaweera*, Chethika Abenayake*⊙, Niroshan Bandara⊙*⊙, Chathura De Silva⊙*⊙

Department of Town & Country Planning, Urban Simulation Laboratory, University of Moratuwa, Moratuwa, Sri Lanka

⊙ These authors contributed equally to this work.
* amilabj@uom.lk (AJ); 162326c@uom.lk, nesharanaweera29@gmail.com (NR); chethika@uom.lk (CA); niroshans@uom.lk (NB); chathurads@uom.lk (CDS)

**Data Availability Statement:** All relevant data are within the paper and its Supporting Information files.

## Abstract

Vegetation land fragmentation has had numerous negative repercussions on sustainable development around the world. Urban planners are currently avidly investigating vegetation land fragmentation due to its effects on sustainable development. The literature has identified a research gap in the development of Artificial Intelligence [AI]-based models to simulate vegetation land fragmentation in urban contexts with multiple affecting elements. As a result, the primary aim of this research is to create an AI-based simulation framework to simulate vegetation land fragmentation in metropolitan settings. The main objective is to use non-linear analysis to identify the factors that contribute to vegetation land fragmentation. The proposed methodology is applied for Western Province, Sri Lanka. Accessibility growth, initial vegetation large patch size, initial vegetation land fragmentation, initial built-up land fragmentation, initial vegetation shape irregularity, initial vegetation circularity, initial building density, and initial vegetation patch association are the main variables used to frame the model among the 20 variables related to patches, corridors, matrix and other. This study created a feed-forward Artificial Neural Network [ANN] using R statistical software to analyze non-linear interactions and their magnitudes. The study likewise utilized WEKA software to create a Decision Tree [DT] modeling framework to explain the effect of variables. According to the ANN olden algorithm, accessibility growth has the maximum importance level [44] between -50 and 50, while DT reveals accessibility growth as the root of the Level of Vegetation Land Fragmentation [LVLF]. Small, irregular, and dispersed vegetation patches are especially vulnerable to fragmentation. As a result, study contributes detech and managing vegetation land fragmentation patterns in urban environments, while opening up vegetation land fragmentation research topics to AI applications.

**Funding:** The authors would like to acknowledge the Senate Research Committee Conference & Publishing Support Grant, University of Moratuwa, Sri Lanka. The funders had no role in study design, data collection and analysis, decision to publish, or preparation of the manuscript.

**Competing interests:** The authors have declared that no competing interests exist.

## 1. Introduction and literature review

Rapid changes in land use and land cover have had a global influence on the environment, economy, and society [1–3]. As a result, for urban planners and experts in charge of land use planning and management, it has become a major concern [4]. Forest, agricultural, marsh, scrub, and green zones are all vegetation land-uses that are threatened by growing urbanization across the world [5,6]. Therefore, the configuration and composition of vegetation lands are changing [7]. According to recent studies, anthropogenic activities have damaged 60 percent of ecological services worldwide [8]. Urban regions are home to 56.2 percent of the world's population [9]. Consequently, fast vegetation cover changes in urban areas can be identified [10]. The primary causes of diminishing vegetative land use are urban growth and sprawl [11,12]. Many studies have used GIS and Remote Sensing tools as well as landscape metrics to assess the extent of diminishing vegetation cover or changes.

Methods including division index, patch density, number of patches, area-weighted index, and others have been used [13,14]. GIS and Remote Sensing-based analytics include the Normalized Difference Vegetation Index [NDVI] [15–17], supervised and unsupervised satellite image classifications for change detection [18–20]. Several AI and machine learning approaches, as well as applications such as MOLUSCE [21] are being used to assess the level of changes in the vegetation cover, methods. FUTURES [22], and the SLEUTH Model [23,24], have been studied to anticipate future land cover changes.

When investigating vegetation cover change, vegetation fragmentation is an important factor to consider [25]. Fragmentation is a landscape ecology term that describes the process of dividing land parcels into smaller ones [26]. Land fragmentation is defined as "a situation where one area/unit is composed of a large number of parcels that are too small for their rational utilization " [27]. Landscape ecology is the interaction of ecological processes with spatial patterns, according to Forman [28]. The "interaction between spatial pattern and ecological process—that is, the causes and effects of spatial variability across a variety of scales" is emphasized in landscape ecological theory [26,28]. The Patch-Corridor-Matrix model outlines the major aspects of a landscape or specific spatial elements [26]. The process of fragmentation, according to landscape ecology, is the changing of patches, corridors, and matrix [26,29]. The Patch-Corridor-Matrix model, which is the essential approach for quantifying vegetation land fragmentation in landscape metrics, is depicted in Fig 1. Fragmentation can be investigated from two perspectives: ecological process and spatial pattern [30].

In that context, this study focused with the spatial implications of vegetation land cover. The study defines vegetation land fragmentation as the process of dividing vegetation land cover into smaller patches or areas because of various anthropogenic activities, as defined by landscape ecology. It also relates to differences in vegetation shape, size, composition, and distribution [29,31]. The key interest here is the division of vegetation patches, not changes in their configuration, and this study regards vegetation configuration and changes as one of the driving causes for vegetation land fragmentation. To quantify vegetation land fragmentation, the Landscape Division Index of landscape metric is utilized. Empirical research indicates the impact of anthropogenic activities on vegetation land fragmentation, particularly in urban areas [32]. In addition, studies point to population density and growth [5,33,34], building density [14,34], decentralization of economic policy [3], infrastructure development, distance to urban centers and specifically transportation development as major causes of vegetation land fragmentation [35,36]. Land is a limited resource, and when there is a greater need for urbanization, people divide or infringe on vegetative areas for urbanization objectives [37,38]. Biodiversity, ecosystem service deterioration, and habitat isolation are all affected by the fragmented structure of vegetation [39,40].

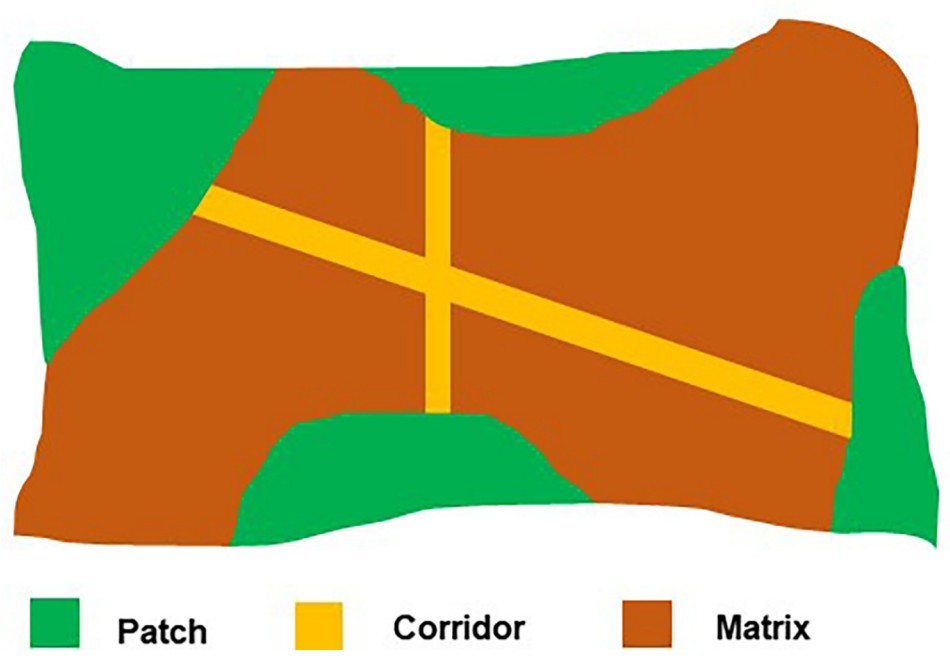

**Fig 1. Patch-corridor-matrix model.**

According to the findings of Abdullah et. al. [41] the five most populous cities in Bangladesh—Dhaka, Chattogram, Khulna, Rajshahi, and Sylhet—have lost more than USD 628 million as a result of specific ecosystem degradation, with the decline of green and blue land areas being the primary cause of this loss. The urban heat island effect is mostly caused by the degradation of vegetated land. For example [42], study demonstrates that due to human limits on agricultural operations, enforcing lockdown following the COVID-19 has increased the amount of vegetative land cover in the Indo-Gangetic basin, India, throughout the winter. This increases the amount of evapotranspiration, which accounts for about half of the region's cooling [42]. Additionally, [43] shows that in five main cities in Bangladesh, including Dhaka, Chittagong, Khulna, Rajshahi, and Sylhet, a lack of green space and a high proportion of impermeable surface are significant contributors to urban heat. Furthermore, vegetation works as a natural flood barrier; hence, when it vanishes or changes as a result of development, there are serious flooding problems in urban areas. According to [44] research, mangroves, marshes, and coastal vegetation are essential for preventing coastal flooding in the world. The irregularity and complexity of fragmented lands can affect land-use efficiency, lowering the economic return on agricultural land [3,45,46]. Also, the urban structure's fragmentation might operate as a barrier to social connection [47]. As a result, vegetation land fragmentation has an impact on sustainability because of the physical structure of land use [1,11]. Therefore, urban planners are now more interested in exploring vegetation land fragmentation alongside landscape and environmental planners [5]. So far, national, regional, and local level empirical studies have been used to study vegetation land fragmentation, while noting the extremely limited yet successful research efforts on vegetation land fragmentation [48]. However, there is an unmet demand for multi-factor modeling of vegetation land fragmentation in urban environments [2,33]. Because vegetation land fragmentation is caused by various variables in a complex environment, the traditional linear regression model is unable to convey the relevance of influencing factors [1,21]. To examine nonlinear interactions in the urban setting, several

recent research has used AI-based modeling tools [1,22,47–49]. However, developing an AI-based modeling framework with multiple factors to explicitly model vegetation land fragmentation is a knowledge gap that needs to be investigated to control vegetation land fragmentation and its consequences [3,27]. Even though numerous researchers have discovered factors that contribute to vegetation land fragmentation, it is critical to understand the types of relationships that exist between vegetation land fragmentation and influencing factors (rules) to make spatial planning judgments. Most of the recent research have identified the variables, but there has been little focus on quantitatively explaining the relationship [50]. To develop efficient land-use regulations and guidelines, it is necessary to determine the magnitude of variables' impacts on vegetation land fragmentation [27,47]. even though successful research attempts on vegetation land fragmentation have focused on forest land cover rather than the entire vegetation land cover since they are more oriented to the ecological domain [51]. In addition, vegetation land fragmentation occurs in both rural and urban areas while generating the same kind of impacts [32,36]. Since the study focuses on the spatial pattern of vegetation land use rather than the ecological process, this study focuses more on vegetation land fragmentation in urban areas.

Sri Lanka is a South Asian country with a diverse range of green and blue land uses; and has become a gateway to Asia as a result of regional development projects—such as Port City Colombo, Hambantota Harbor, and Airport Development. Rapid infrastructure development projects, such as expressway construction and regional infrastructure development, have resulted in a reduction in the percentage of land covered by vegetation in Sri Lanka [52]. Western Province in Sri Lanka has seen significant urbanization and has lost most of its natural cover [14,53]. Various research has used GIS and remote sensing techniques such as NDVI [54–56], Satellite image classifications, and spatial metrics to explore land cover change in Sri Lanka [57,58]. Some researchers have even used AI-based applications such as the SLEUTH model at the regional level to anticipate land cover changes [59]. However, a handful of research have looked into vegetation fragmentation in urban areas, in Sri Lanka. In addition, one study has focused solely on paddy field fragmentation and its economic consequences [46]. Because Sri Lanka is rich in biodiversity and forest, wetlands and other vegetation covers, it is important to investigate the level of vegetation fragmentation. Due to the growing urbanization patterns in the Western Region, this research focuses on vegetation fragmentation in the Western Province; because studies of vegetation fragmentation are necessary for determining implications for reducing fragmentation's negative effects and ensuring sustainable development in urban areas [4,27,60]. Urban planning agencies such as the Urban Development Authority, the National Physical Planning Department, and the Central Environmental Authority, as well as local governments, require tools and model applications to forecast future scenarios in vegetation fragmentation and identify their factors in order to develop effective policies to control the level of fragmentation in Sri Lanka and to make society more sustainable. Although there are already developed AI-based applications such as SLEUTH, MOLUSCE, they are oriented to analyze the urban growth or land cover changes. Therefore, it is essential to develop an AI-based model framework to model vegetation land fragmentation not only in Sri Lanka but also for other countries.

The objective of this study is to develop an AI-based simulation framework to simulate vegetation land fragmentation in urban environments. The non-linear relationships and magnitude of impacting elements were identified using a feed-forward ANN in this study. Under the AI modeling technique, the study also built a DT modeling framework to explain the effect of many variables on vegetation land fragmentation. The study exclusively used data from Sri Lanka's Western Province to create the model. As a result, the study's scope is restricted to the Western Province. This work, on the other hand, might be viewed as the first attempt to

measure and simulate vegetation fragmentation in the Sri Lankan environment. Furthermore, the methodological scientific relevance and contribution of AI-based modeling approaches adds to the incentive to this paper.

## 2. Materials and methods

### 2.1. Study area and data sources

Western Province, Sri Lanka, was chosen as the case study considering data availability and the capacity to meet the sample size requirements of AI models. According to the 2012 census and statistics report in Sri Lanka, the Western Province covers 3684 km$^2$ and has a population of 5.85 million people. It has the densest urban population and covers 46 percent of the total built-up area. With significant urban and infrastructural developmental projects such as expressways, port and airport extensions, and power stations, the Western Region serves as Sri Lanka's commercial hub. It also contains 2505 Grama Niladhari Divisions (GND), which are local administrative borders, and these 2505 GNDs were utilized as the AI model's sample in this study.

For the study, primary and secondary data sources were employed. The vegetation and built-up land-uses in the Western Province were mostly extracted from USGS Lands at 7 satellite images in 2000 and 2010. The elements of vegetation land fragmentation were calculated using data from the Survey Department's Road network and contour layers, as well as data from the Census and Statistic Department of Sri Lanka's population and building units. Table 1 shows the data sources for each study variable in further detail.

### 2.2. Method

The study is based on a review of the literature, which included both theoretical and empirical investigations, to determine the factors that cause vegetation land fragmentation from the beginning [Fig 2]. The identified factors were then conceptualized as patch, corridor, matrix, and other factors based on the theoretical explanation of landscape ecology. Fig 3 illustrates the conceptual framework for vegetation land fragmentation in this study. The study relied on 20 different parameters, which are included in Table 1, and used USGS satellite images of Western Province in 2000 and 2010 to classify them into vegetation and built-up classifications (with over 75% classification accuracy). At the GND level, each factor is calculated using geoprocessing tools available in ArcMap and QGIS. Even though the components were conceived as patch, corridor, matrix, and other categories in the study, the geographic entity employed was GND, Sri Lanka's smallest administrative entity. The availability of secondary data was also taken into account in this investigation. As a result, the model outputs are also tied to GND administrative borders rather than spatial features such as patches, corridors, or matrix. The vegetation fragmentation was quantified using FRAGSTATS (Landscape Division Index). During the database development, existing local reports determined some of the quantitative parameters. Before defining the models, Principal Components Analysis was used to exclude the multi-correlated and least-correlated factors. After all, the study used 1750 training data and 750 testing data to design and validate the ANN and DT models. Finally, using the ANN, it calculated the future LVLF in Western Province. The next sections will provide extensive explanations for each process.

Accessibility Growth (AG), Initial Vegetation Large Patch Size (IVLPS)], Initial Vegetation Land Fragmentation (IVLF), Initial Built-Up Land Fragmentation (IBLF)], Initial Vegetation Shape Irregularity (IVSI), Initial Vegetation Circularity (IVC), Initial Building Density (IBD), and Initial Vegetation Patch Association (IVPA) are among the eight input variables after exclusion of least correlated and multi correlated factors. According to the study, 'initial' refers

**Table 1. Factors of vegetation land fragmentation.**

| Category | Name of the variable | Definition/Equation | Data Source |
|---|---|---|---|
| Patch | Level of vegetation land fragmentation (LVLF) (Dependent variable) | Landscape division index = DIVISION [61]<br>LVLF = (Existing vegetation fragmentation–Initial vegetation fragmentation) / Initial vegetation fragmentation | Landsat 7 Satellite images (USGS)] |
| | Vegetation Fragmentation (VF) | Landscape division index = DIVISION [61] | Landsat 7 Satellite images (USGS) |
| | Large Patch Size | Largest patch index = LPI (62) | Landsat 7 Satellite images (USGS) |
| | Patch association | Euclidean Nearest-Neighbor Distance = ENN [61] | Landsat 7 Satellite images [USGS] |
| | Shape irregularity | Area-Weighted Mean Shape Index = AWMSI [61] | Landsat 7 Satellite images (USGS) |
| | Shape circularity | Related Circumscribing Circle = CIRCLE [61] | Landsat 7 Satellite images (USGS) |
| Corridor | Accessibility growth [AG] | Growth of closeness centrality and Betweenness centrality<br>AG = (Existing accessibility–Initial accessibility)/Initial accessibility | Survey Department, Sri Lanka |
| | Accessibility | Accessibility = $\sqrt{}$ Closeness centrality * Betweenness centrality | Survey Department, Sri Lanka |
| | Closeness centrality | Average shortest distance to all other nodes [62] | Survey Department, Sri Lanka |
| | Betweenness centrality | Degree of nodes standing between each other [62] | Survey Department, Sri Lanka |
| | Road density | RD = Area of roads/ Area of GND | Survey Department, Sri Lanka |
| | Growth of road density | RDG = (Existing road density- Initial road density)/Initial road density | Survey Department, Sri Lanka |
| Matrix | Built-up fragmentation | The number of buildings within the built-up land extent.<br>Built-up fragmentation = Number of buildings / Area of built-up | -Satellite images (USGS) for the built-up area<br>-Number of the buildings by Census and statistics |
| | Building density growth [BDG] | BDG = (Existing building density–Initial building density)/ Initial building density | Landsat 7 Satellite images (USGS) |
| | Building density | Building density = Area of built-up / Area of GND | Landsat 7 Satellite images (USGS) |
| | Population density | PD = Number of population / Area of GND | Department of Census and Statistics, Sri Lanka |
| | Population growth | PG = (Existing population–Initial population) / Initial population | Department of Census and Statistics, Sri Lanka |
| Other | Access to urban activities | Ability to access urban activities | [63] |
| | Infrastructure availability | Availability of water and electricity<br>Infrastructure availability = $\sqrt{}$ % of water availability * % of electricity availability | [63] |
| | Slope | Mean slope of the GND | Survey Department, Sri Lanka |
| | Elevation | Mean elevation of the GND | Survey Department, Sri Lanka |

to the year 2000. The spatial distribution of each variable is presented in Fig 4. The classification levels are indicated in a 1–5 scale (very low to very high). Table 2 indicates the numerical values of each variable under classifications shown in Fig 4.

**2.2.1. Landscape division index.** The landscape division index was employed to calculate the vegetation land fragmentation during the database development stage in the research. The pace of change in vegetation land fragmentation over time is referred to as the LVLF. VLF represents the term of vegetation land fragmentation in the equation.

$$LVLF = \frac{VLF\ of\ year2 - VLF\ of\ year1}{VLF\ of\ year1} \qquad (1)$$

The landscape division index was used to calculate the LVLF at the class level. In the FRAGSTATS software, it is one of the landscape metrics. By detecting random pixels in each area,

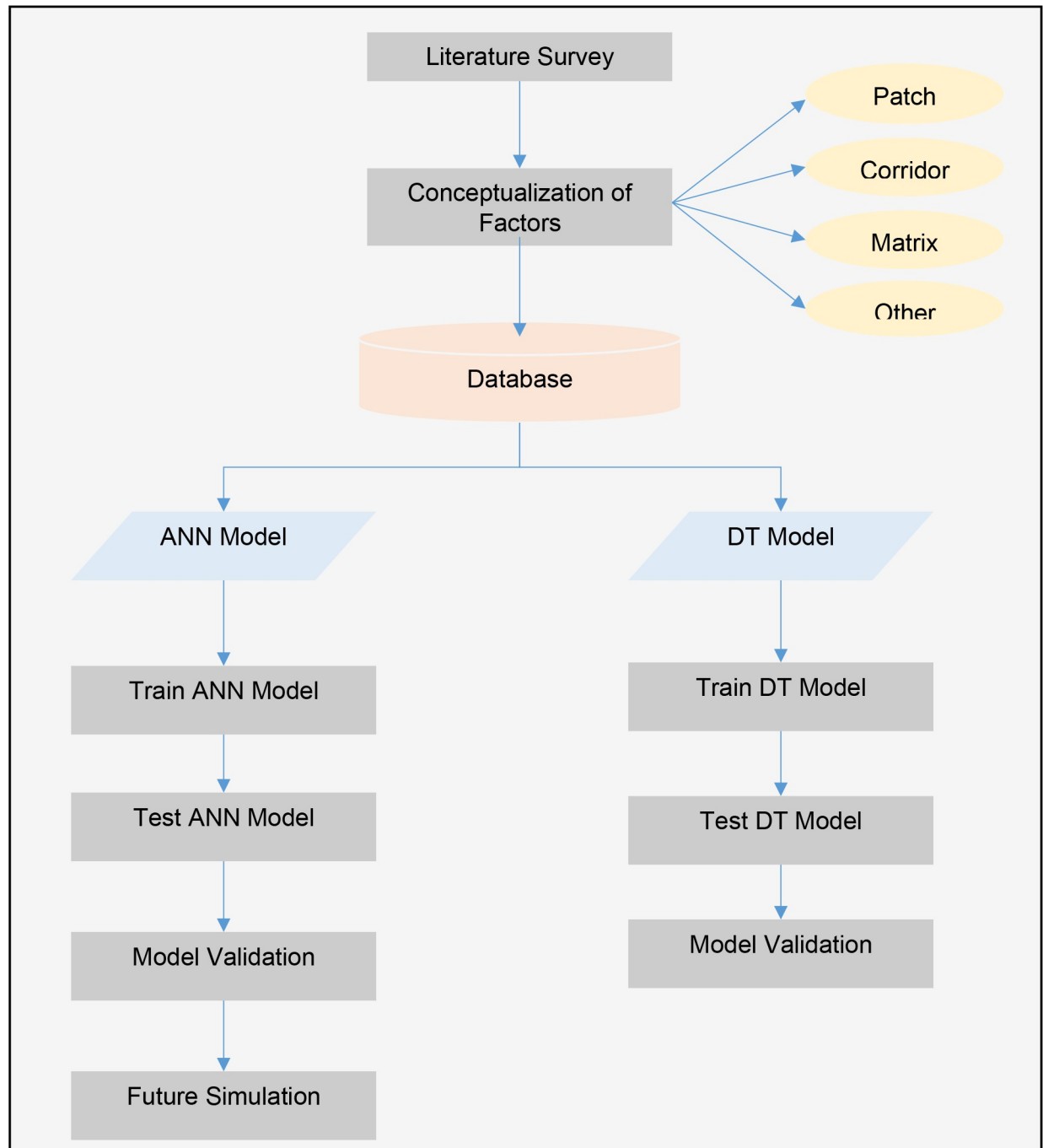

**Fig 2. Steps of research methodology.**

the landscape division index assesses the division of the same land use category. It determines the likelihood of two random pixels not being in the same patch [64].

$$Landscape\ Division\ Index = [1 - \sum_{i=1}^{n} * \sum_{j=1}^{n} [\frac{a_{ij}^2}{A}]] \quad (2)$$

The complete landscape area is denoted by the letter $A$ [61]. $aij$ is the area of patch $ij$ [61].

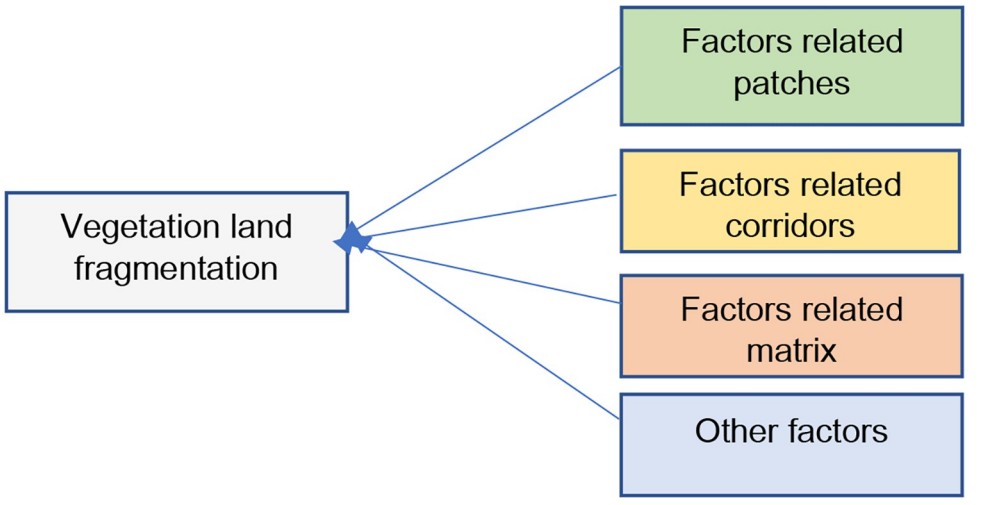

**Fig 3. Conceptual framework.**

**Fig 4. Spatial distribution of factors.**

**Table 2. Numeric values of variables' categories.**

| Type of variable | Category | Name of variable | Classification levels |
|---|---|---|---|
| Dependent variable | | LVLF | <20%- Very low |
| | | | 20%-40%- Low |
| | | | 40%-60%—Moderate |
| | | | 60% -80%—High |
| | | | >80%- Very high |
| Independent variables | Patch | IVLPS | <20- Very low |
| | | | 20-40- Low |
| | | | 40–60—Moderate |
| | | | 60–80—High |
| | | | >80- Very high |
| | | IVPA | <50m-Very low |
| | | | 50m-100m- Low |
| | | | 100m-150m- Moderate |
| | | | 150m-200m- High |
| | | | >200m- Very high |
| | | IVLF | <0.2- Very low |
| | | | 0.2–0.4- Low |
| | | | 0.4–0.6- Moderate |
| | | | 0.6–0.8- High |
| | | | 0.8–1.0- Very high |
| | | IVSI | <1.5- Very low |
| | | | 1.5–2.0- Low |
| | | | 2.0–2.5- Moderate |
| | | | 2.5–3.0- High |
| | | | >3.0 -Very high |
| | | IVC | <0.2- Very low [Circle] |
| | | | 0.2–0.4- Low |
| | | | 0.4–0.6- Moderate |
| | | | 0.6–0.8- High |
| | | | 0.8–1.0- Very high [Linear] |
| | Corridor | AG | <20%- Very low |
| | | | 20%-40%- Low |
| | | | 40%-60%—Moderate |
| | | | 60% -80%—High |
| | | | >80%- Very high |
| | Matrix | IBLF | <0.2- Very low |
| | | | 0.2–0.4- Low |
| | | | 0.4–0.6- Moderate |
| | | | 0.6–0.8- High |
| | | | 0.8–1.0- Very high |
| | | IBD | <0.2- Very low |
| | | | 0.2–0.4- Low |
| | | | 0.4–0.6- Moderate |
| | | | 0.6–0.8- High |
| | | | >0.8- Very high |

The landscape division index has a value ranging from 0 to 1 [61]. The value 0 indicates that the area is not fragmented. When the value is close to 1, the land is highly fragmented. The value will be 0 if $j$ and $i$ pixels are in the same patch [61]. If the pixels $j$ and $i$ are in distinct patches, the value ranges from 0 to 1 [61]. The division value will be closer to 1 if $i$ and $j$ pixels are located inside smaller areas.

**2.2.2. AI-based modeling frameworks.** The complex urban environment processes and challenges have been modeled using AI-based modeling frameworks [65]. The bulk of research attempts to do away with the Linear Regression model, since it is insufficient when dealing with a complex environmental issue [66]. The advantages of using AI-based modeling frameworks include the ability to model with several variables and the ability to identify the cluster influence of the variables [67]. Consequently, AI models can be used to identify nonlinear connections. To meet the research objectives, this study used both ANN and DT.

*2.2.2.1. Artificial Neural Network [ANN].* In this study, the first AI-based approach was ANN, which was used to predict vegetation fragmentation. The approach of an intelligent system that imitates the behavior of the human brain [neurons] is known as ANN [68]. The first mathematical neural model was established in 1943 by Warren McCulloch and Walter Pitts [69]. Input, hidden, and output layers [70]are the three primary layers of an ANN, as shown in Fig 5. Fig 6 represents a rudimentary neural network and its output-generating function. The input variables are X1, X2, and X3, and the weights of respective inputs are W1, W2, and W3. The bias node is 'b,' and the output is 'Y.'. The activation function, which is the neural network's mapping process, is utilized to smooth the output results [67]. As a result, the output function is the multiplication of weight, input, and bias [69].

There are two types of neural networks: supervised and unsupervised [70]. This research uses a supervised neural network to train the algorithm from provided outputs and inputs by

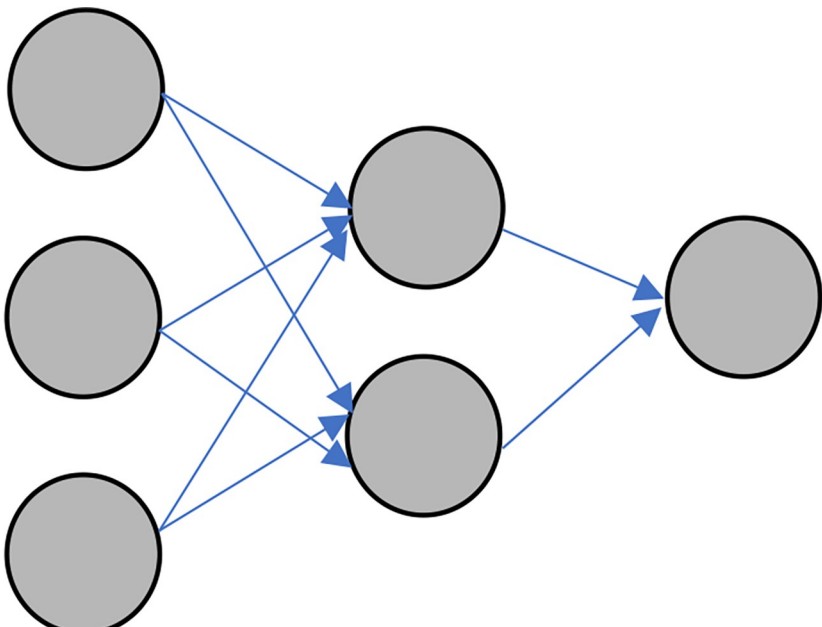

Input Layer Hidden Layer Output Layer

**Fig 5. Layers of neural network.**

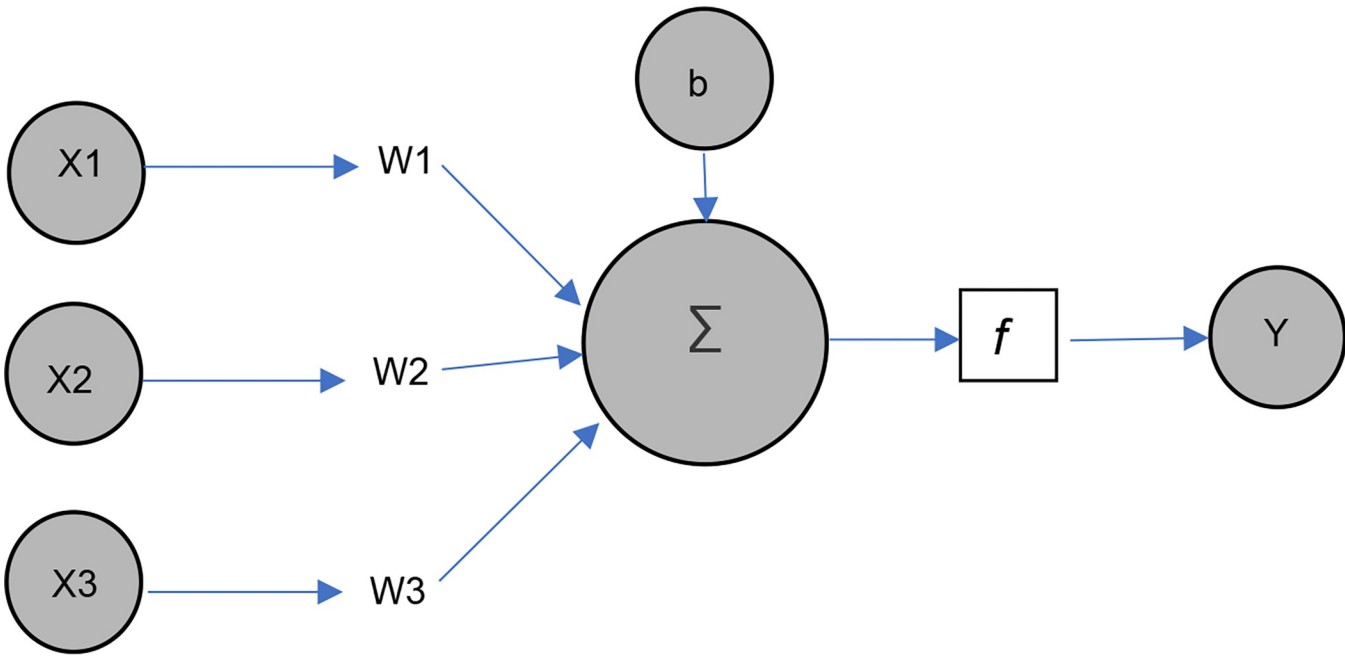

**Fig 6. Simple neural network.**

changing the predicted output to the actual output while updating the weights according to neural network rules. Data categorization, change detection, grouping, forecasting, and predictions are all possible using supervised neural networks (Deep Learning) [71]. It uses hidden layers to train the existing nonlinearity between input and output variables and predicts the output [72]. Therefore, it can determine the level of significance of input variables in forecasts [68]. It is more useful for simulating spatial dynamics and identifying changes in variables [73].

$$Y = \sum (w * x) + b \tag{3}$$

$$y = \frac{x - xmin}{xmax - xmin} \tag{4}$$

Before the training process, the data was standardized using the lowest and maximum values. $x$ is the original data value, and $y$ is the matching data value after the maximum and lowest values of $x$ have been normalized [74]. The major packages used in R Studio for ANN preparation are Keras, Neuralnet, and Tensorflow. The default algorithm of the R language's rprop + algorithm, which is based on resilient backpropagation, was utilized in the study. To identify the minimal error function, it changed the weights of input nodes [70]. Because the neural network's linear output function is false, the default logistic activation function was used, with output values ranging from 0 to 1. Mean Square Error [MSE] [75] and Root Mean Square Error [RMSE] [76], which examine the error between predicted and tested data, were used to evaluate the model [70].

$$MSE = \frac{1}{n} \sum_{i=1}^{n} [Predicted - Actual]^2 \tag{5}$$

$$RMSE = \sqrt{\sum_{i=1}^{n}[\frac{[Predicted - Actual]^2}{n}} \tag{6}$$

Using the R neuralnet package's Olden and Lekprofile algorithms, the study determined the feature significance of variables and rules. Olden is an algorithm that uses the raw value of input-hidden and connection weights of hidden outputs of each input and neuron to indicate the importance of variables [77]. The total of all hidden neurons is then calculated. Each variable's negative and positive importance levels can be displayed by the olden algorithm [78]. In the developed ANN, the lekprofile method is used to verify the sensitivity of each variable to the independent variable [68]. Other factors are kept constant while determining a variable's sensitivity. For each variable, the Lekprofile algorithm generates a graph with probable sensitivities or relationship curves. The complete technique of framing the ANN model through R is shown in Fig 7.

*2.2.2.2. Decision Tree (DT).* In machine learning, the DT is a strong classification and modeling technique [51] which is employed as the second AI-based technique in this study. It builds a tree structure similar to a flow chart and is classified as supervised machine learning [79]. Individuals' behavior or decisions about one another can be predicted by the DT [80]. The DT was calculated using WEKA software in this investigation. The technique utilized was the J48 algorithm, which solely classified the model's significant factors. The data were first normalized using the equation of minimum and maximum, and then the arff. file was generated. The study classified normalized data as Very High [VH], High [H], Moderate [M], Low [L], and Very Low [VL] to make it more understandable. The Kappa Statistics [K] [81], Relative Absolute Error [RAE] [82], and Root Relative Squared Error [RRSE] [82] were used to validate the DT model. Fig 8 describes the formulation and validation procedure for DT models.

$$K = \frac{P_{agree} - P_{chance}}{1 - P_{chance}} \tag{7}$$

$P_{agree}$ is about the observed agreement [81]. $P_{chance}$ is agreement expected by chance alone [81]. More than 70% kappa statistic value would represent the validated model in terms of higher accuracy [81].

$$RAE = |\frac{Observed\ Actual\ Value - Expected\ value}{Expected\ Value}|.100\% \tag{8}$$

RAE examines the absolute error to actual value ratio [82]. The discrepancy between the actual value and the measured value is known as absolute error [82]. The model is considered accurate if the RAE is less than 20%.

$$RRSE = \sqrt{\frac{\sum_{j=1}^{n}[P_{ij} - T_j]^2}{\sum_{j=1}^{n}[T_j - \bar{T}]^2}} \tag{9}$$

$$\bar{T} = \frac{1}{n}\sum_{j=1}^{n}T_j \tag{10}$$

The difference between the predicted (*Pij*) and the target value is also calculated by RRSE (*Tj*) [82]. It divides the total squared error of the simple predictor to normalize the total squared error.

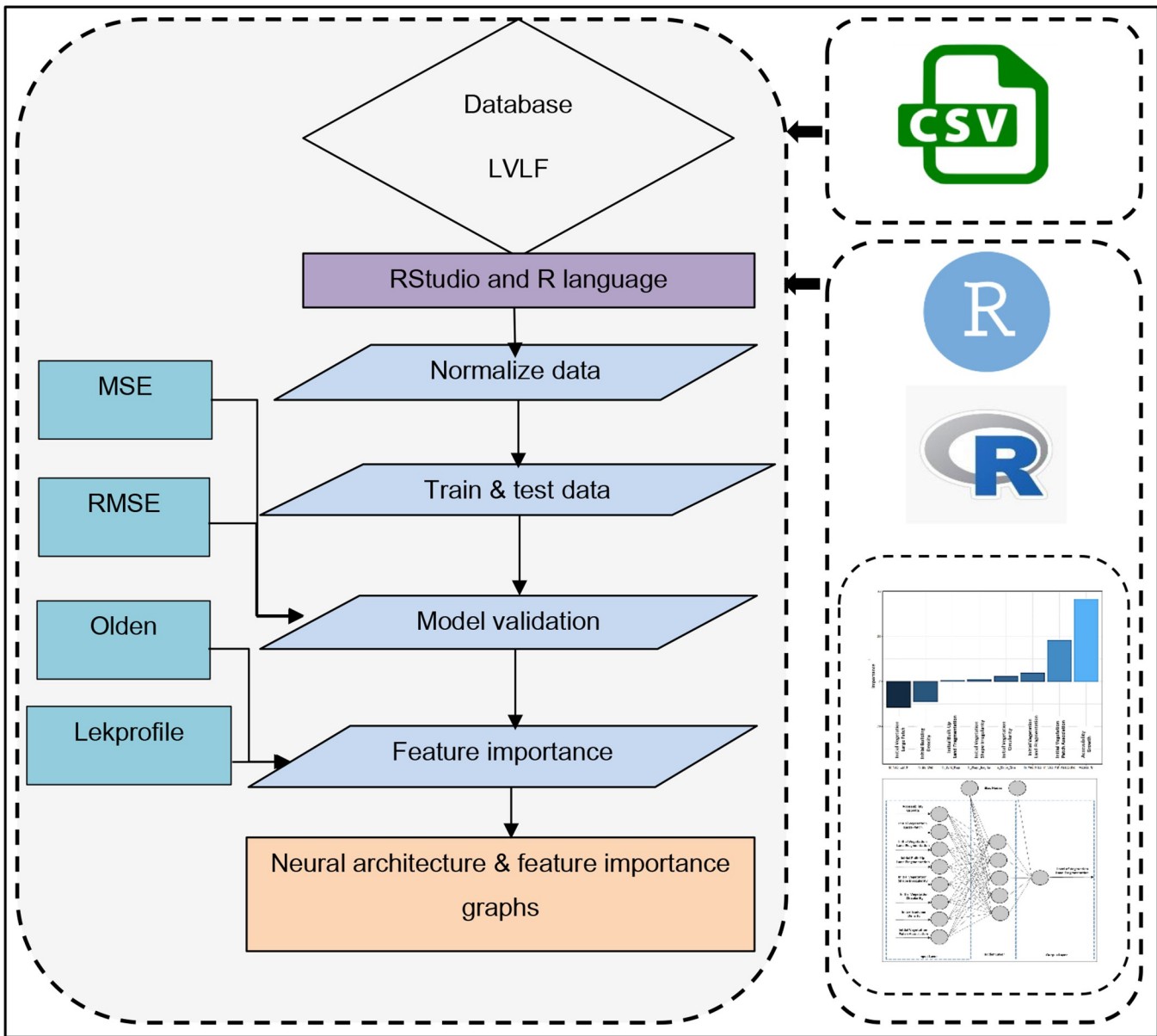

**Fig 7. Computation process of ANN.**

## 3. Results

The study first measured the LVLF and factors of LVLF, then framed the ANN and DT models to discover the variables of LVLF and their relationships, as indicated in the method. Finally, the study used ANN to simulate future LVLF in Sri Lanka's Western Province from 2010 to 2030. Therefore, the results of the ANN and DT models, as well as the simulation results, will be explained in the following section.

### 3.1. ANN

The architecture of the neural network is depicted in Fig 9. Accessibility Growth (AG), Initial Vegetation Large Patch Size (IVLPS), Initial Vegetation Land Fragmentation (IVLF), Initial

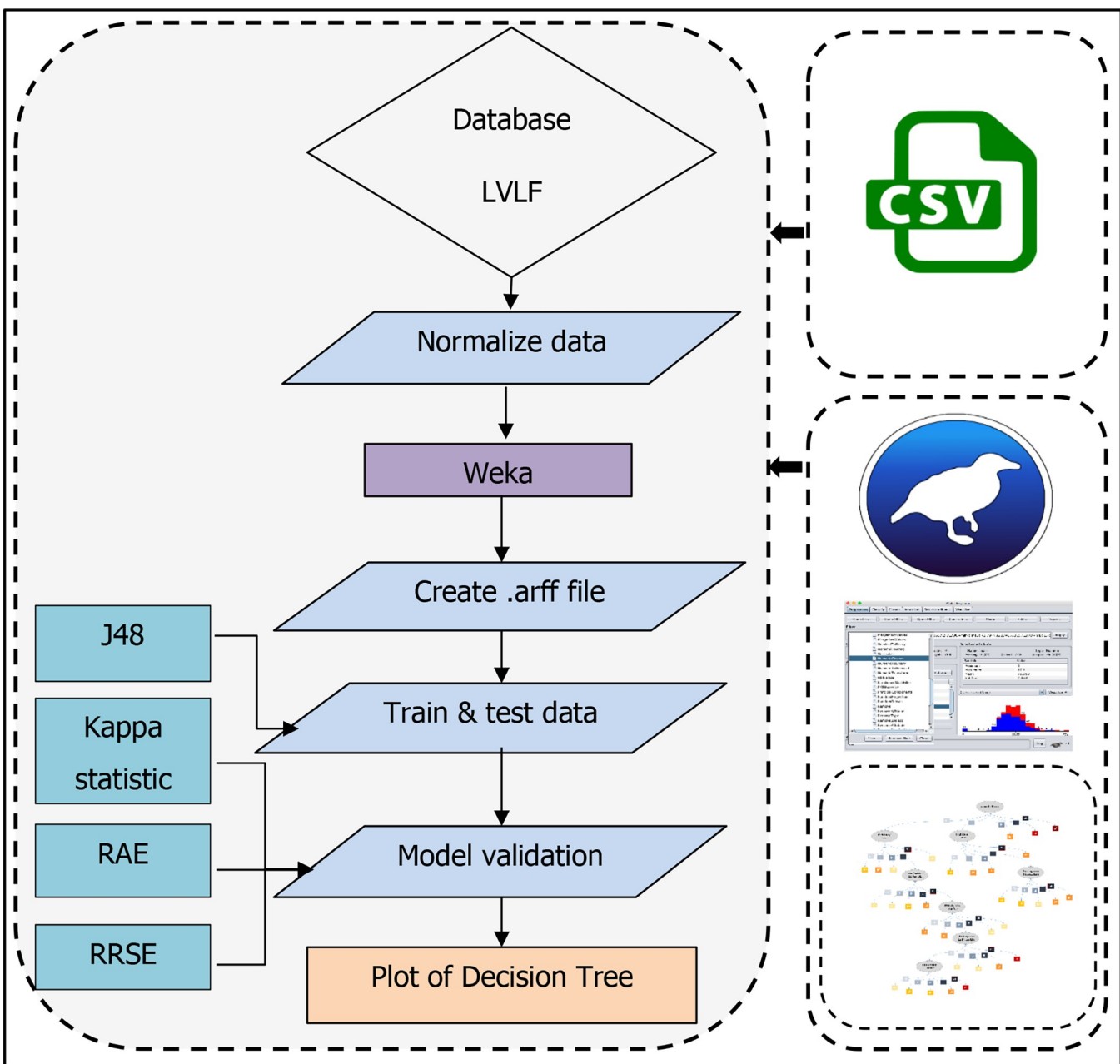

**Fig 8. Computation process of DT.**

Built-Up Land Fragmentation (IBLF), Initial Vegetation Shape Irregularity (IVSI), Initial Vegetation Circularity (IVC), Initial Building Density (IBD), and Initial Vegetation Patch Association (IVPA) are among the eight input variables after the exclusion of least correlated and multi correlated factors. According to the study, the initial refers to the year 2000. LVLF is the model's output variable or dependent variable. There are 5 hidden nodes and 2 bias nodes in the model. The model was trained with 1750 training data and 750 testing data. The MSE and RMSE values are 0.025 percent and 1.574 percent, respectively, indicating that the model accuracy is high; because the output value range is 0 to 1.

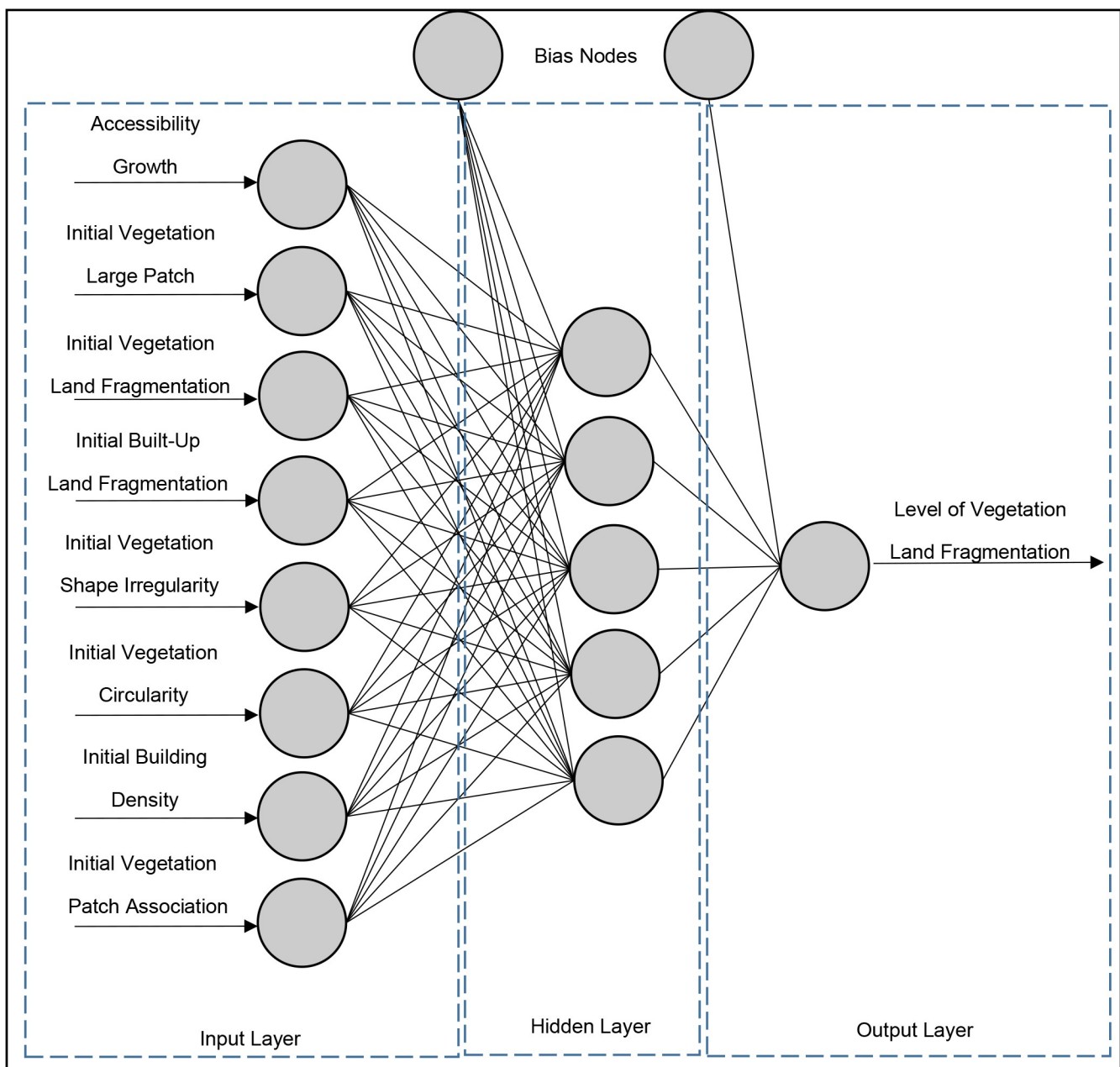

**Fig 9. ANN architecture of LVLF model.**

AG is the positively most significant factor of the LVLF, according to the olden algorithm of ANN (Fig 10). Between the ranges of 50 and -50, the significance level of AG is 45. The IVPA (distance between patches) is the second-most important positive factor which is 23. IVLPS has the greatest negative importance level -14, whereas IBD has the second-highest negative importance level which is -11. The remaining components have negligible positive significance ratings. According to the findings, AG is the most important factor in corridor considerations, and it is also the highest. IVPA and IVLPS are two patch-related parameters that are significantly important.

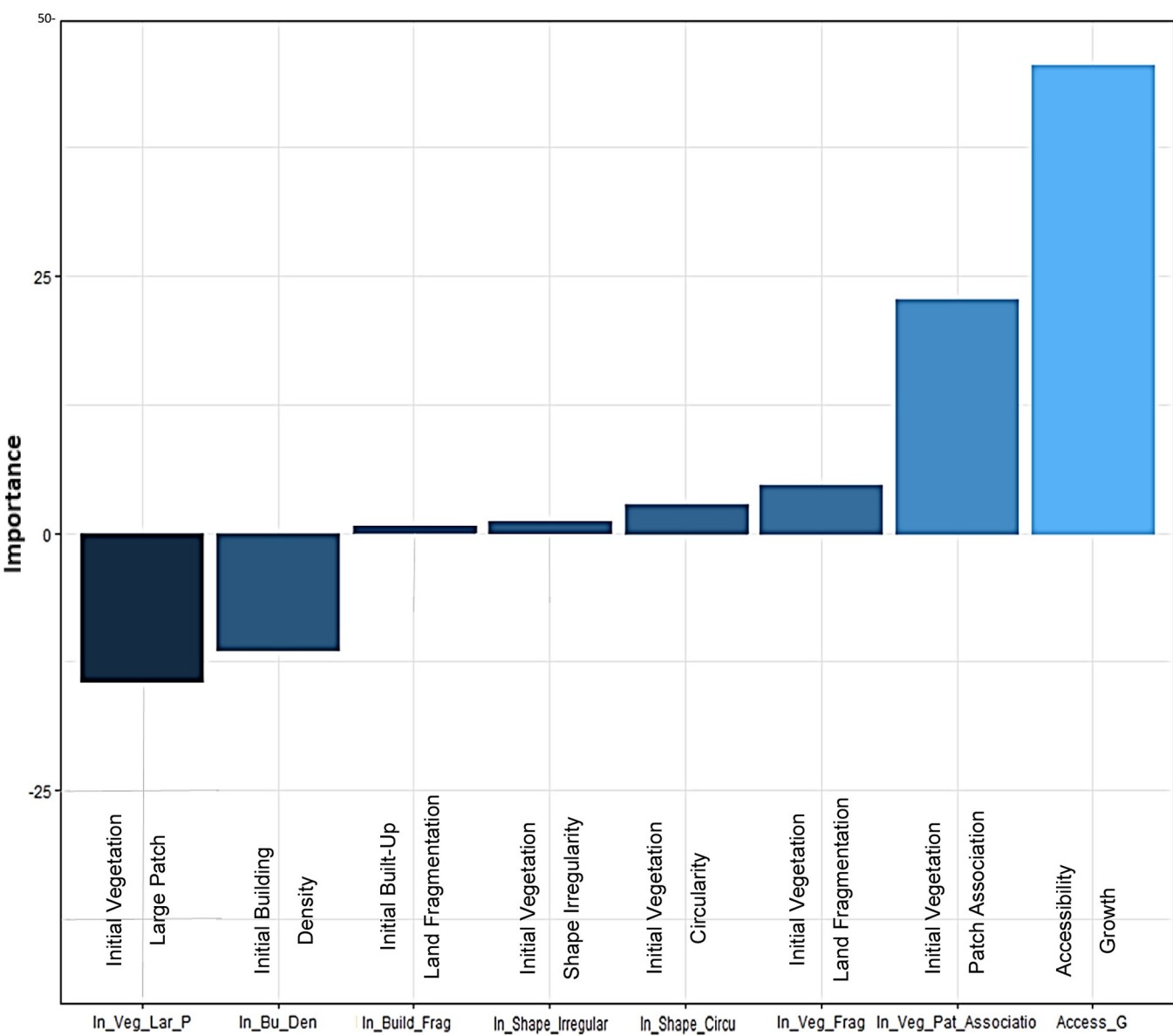

**Fig 10. Results of the olden algorithm: Importance level of factors.**

The rules of each explanatory variable may be determined using ANN's 'lekprofile' analysis. Fig 11 illustrates a composite representation of the LVLF's rules. AG interferes favorably with LVLF, whereas increasing AG causes LVLF to rise dramatically. The relationship between IBD and LVLF is negative, indicating that the two variables do not move in the same direction. With increasing building density, LVLF eventually diminishes. The IBLF and the LVLF have a positive relationship. The LVLF gradually increases as the LBLF grows. When the shape of the vegetation becomes linear patches, LVLF continuously rises, indicating that IVC and LVLF have a positive connection. The IVSI and the LVLF have a positive correlation. LVLF steadily rises as the shape of the vegetation patches becomes more uneven. If the relationship between IVLF and LVLF is positive and LVLF is rising in lockstep with IVLF, it is a positive indicator. IVLPS has a negative relationship with LVLF, and LVLF gradually diminishes as the size of the

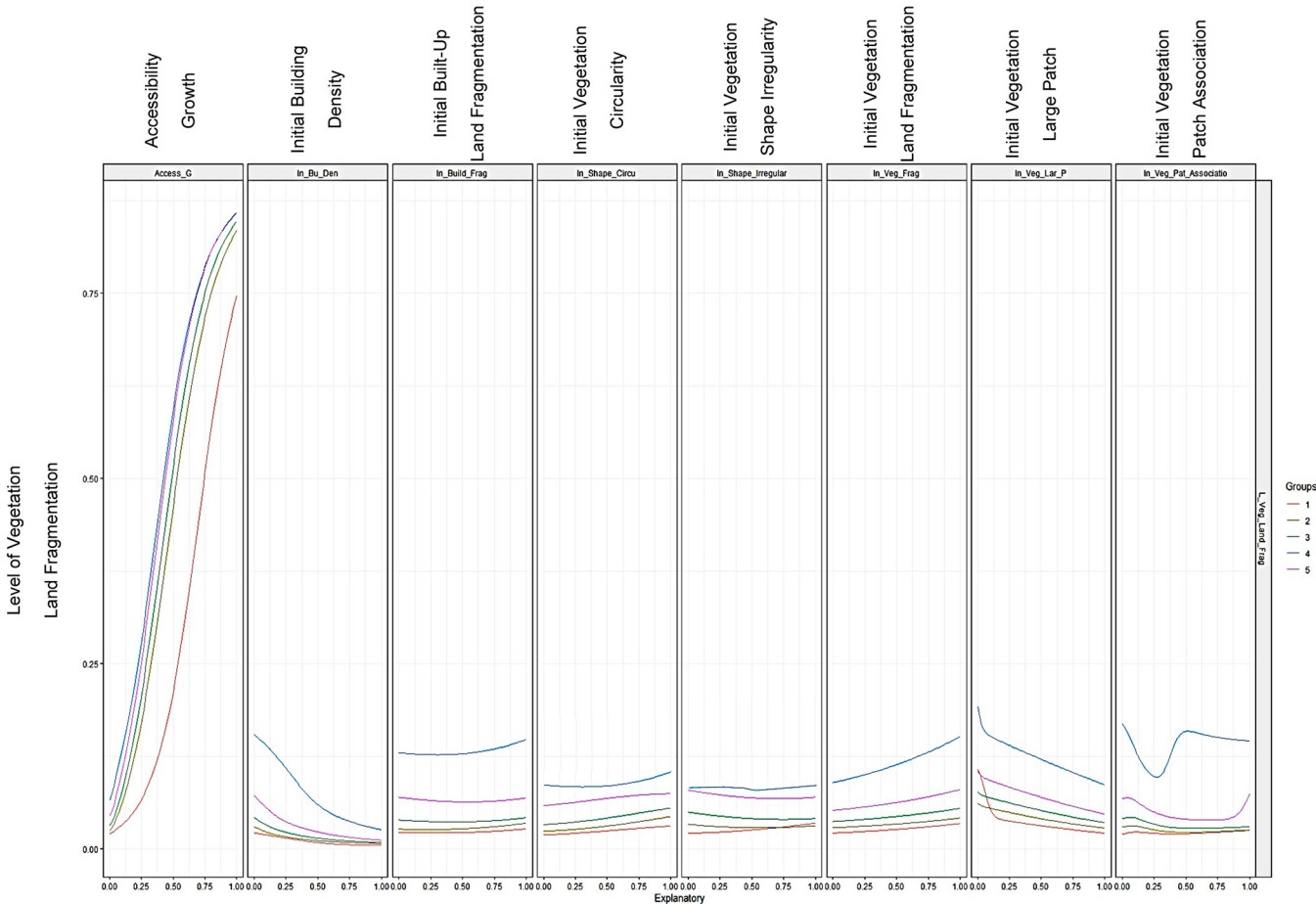

**Fig 11. Rules of variables according to lekprofile algorithm.**

vegetation patch grows larger. The LVLF and the IVPA (distance between patches) have a positive relationship, meaning that increasing the distance between vegetation patches gradually raises the LVLF.

### 3.2. Decision Tree (DT)

The results of the DT model can be used to determine the scenarios of LVLF (Fig 12) and its factors. The DT analyses the link between variables and various scenarios of LVLF with the significant factors as discovered factors in the neural network. The ANN model's greatest priority level factor is AG and the DT model double-proves AG as the root of LVLF. It explains why, in each location, LVLF is particularly high when AG is very high. If the AG is low and the IBD is high, the LVLF is mild. When AG is low and IBD is high, the model uses the IVPA (distance between patches) variable as a branch to explain the various scenarios. If IVPA is high in comparison to earlier connections, LVLF will be moderate. With IVLPS linking to the extremely low AG, high IBD, and very high IVPA, the model has formed a new branch. In keeping with the foregoing relationships, if IVLPS is very high, the LVLF is very low. With the IVLF variable and the IVC variable, another alternative possibility has emerged. In keeping with the connections, if the IVLF is very high, the LVLF is also quite high. With the link of preceding factors, if IVC is very high, LVLF is also very high. Therefore, the DT is a model that can be used to

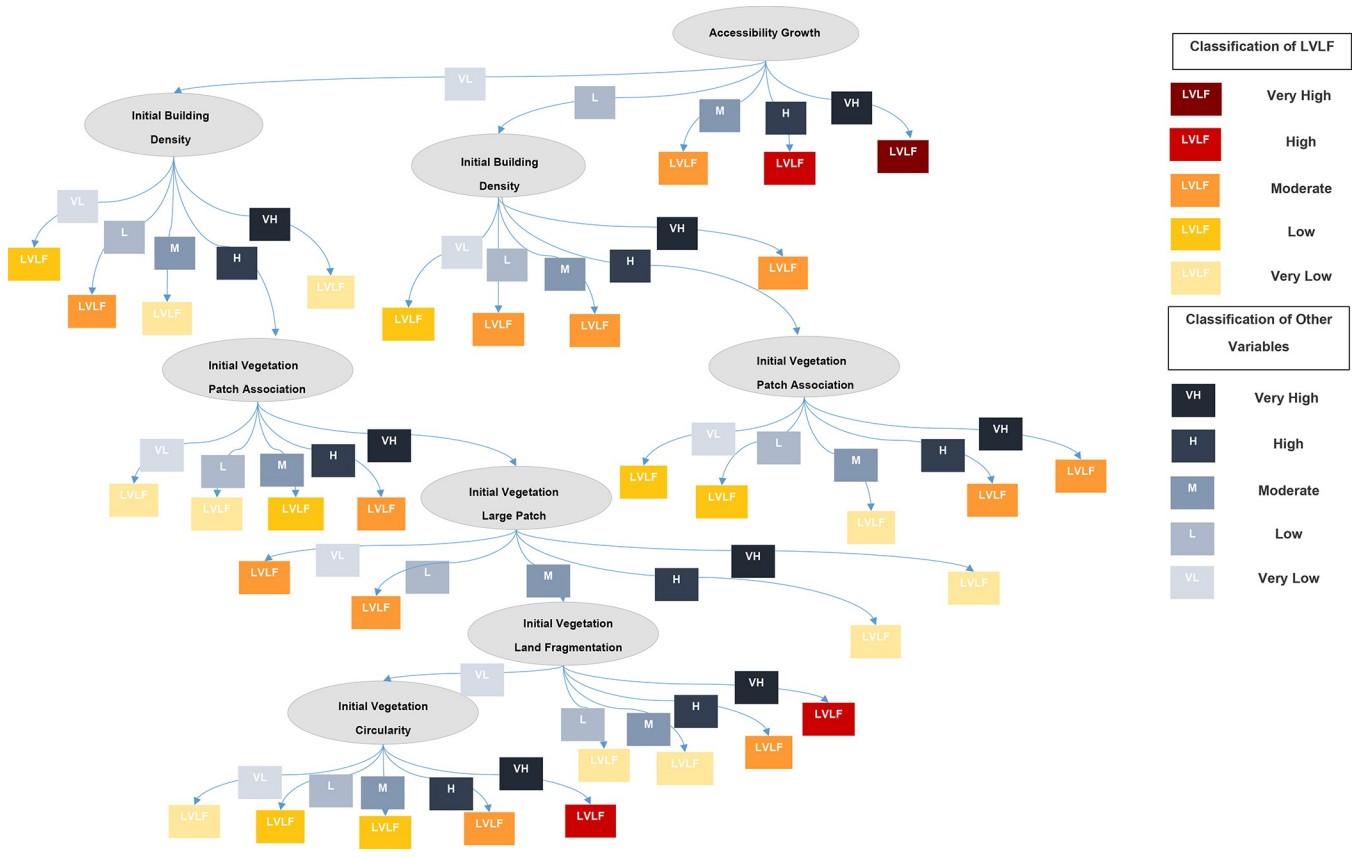

**Fig 12. DT results of the LVLF model.**

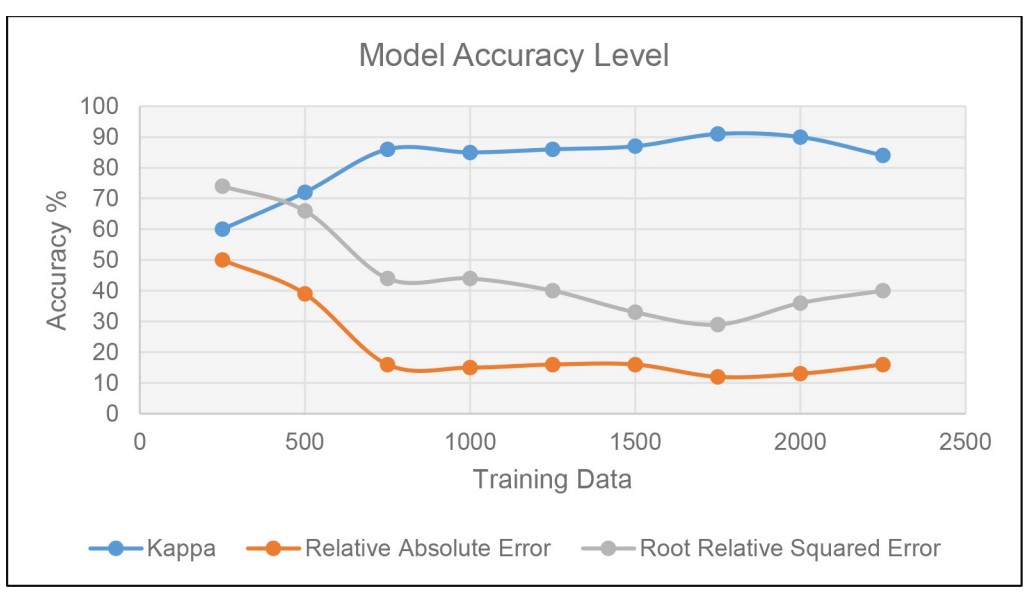

**Fig 13. Validation results based on training data.**

**Table 3. Confusion matrix of DT.**

| | | LVLF | | | | |
|---|---|---|---|---|---|---|
| | | VL | L | M | H | VH |
| LVLF | VL | 168 | 3 | 0 | 0 | 0 |
| | L | 26 | 241 | 0 | 0 | 0 |
| | M | 0 | 2 | 42 | 0 | 0 |
| | H | 0 | 0 | 0 | 181 | 0 |
| | VH | 0 | 0 | 0 | 1 | 86 |

determine LVLF using these variables' connections (factors). The numeric values of each variable related to very low to very high are shown in Table 2.

The results of the model validation demonstrate that 98 percent of the data were properly categorized. RAE and RRSE are 12 and 29 percent respectively. As a result, the model has a greater level of accuracy in terms of validation methods. Fig 13 indicates the percentage of accuracy levels in the tested model using training data. It demonstrates that a minimum of 1000 observations is required to anticipate accurate results. The model accuracy improves as the number of training observations increases from 1000 to 1750. The confusion matrix (Table 3) shows how effectively the DT model classifies the various levels of categories inside the model. Significantly, all the cases with very high LVLF have been accurately identified, whilst the remainder of the groups contain slight misclassifications.

### 3.3. Future simulation

In comparison to the other two districts in the Western Province, the future simulation from 2010 to 2030 (Fig 14) shows a large increase in LVLF in the Gampaha District. It also shows the increase in LVLF along Western Province's expressways and their interchanges. In the Colombo core region, LVLF exhibits a downward trend from 2000 to 2010. The Gampaha District and the Colombo outskirts, on the other hand, have seen a major increase in LVLF.

## 4. Discussion

The ANN and DT models were used to simulate vegetation land fragmentation and for identifying important factors and non-linear relationships. A supervised feedforward neural network [Deep Learning] was employed to investigate the variables of vegetation land fragmentation and their behavior, model future vegetation land fragmentation. The DT [supervised classification] model was employed in identifying possible scenarios in vegetation land fragmentation.

The findings of the ANN's olden and lekprofile algorithms, as well as the DT's J48 algorithm, were stated to meet the research objectives. According to the ANN olden algorithm, AG verifies the maximum importance level 45 between the ranges of -50 and 50. When compared to the recent research covered in this study, this is a surprising discovery because many studies regard accessibility as a road network [83,84].

Further the study calculated accessibility using the centrality measures of betweenness centrality and closeness centrality, which is one of the most recent ways of assessing accessibility using space syntax [85]. The rapid road development in the suburbs, such as expressway extensions, is the explanation for this discovery. The expansion of the road network improves accessibility in both urban and suburban areas [85,86]. Consequently, the model clearly addresses the increase in accessibility expansion and its beneficial effects on vegetation land fragmentation. Urban sprawl has a direct influence on the division of vegetation areas, the study's

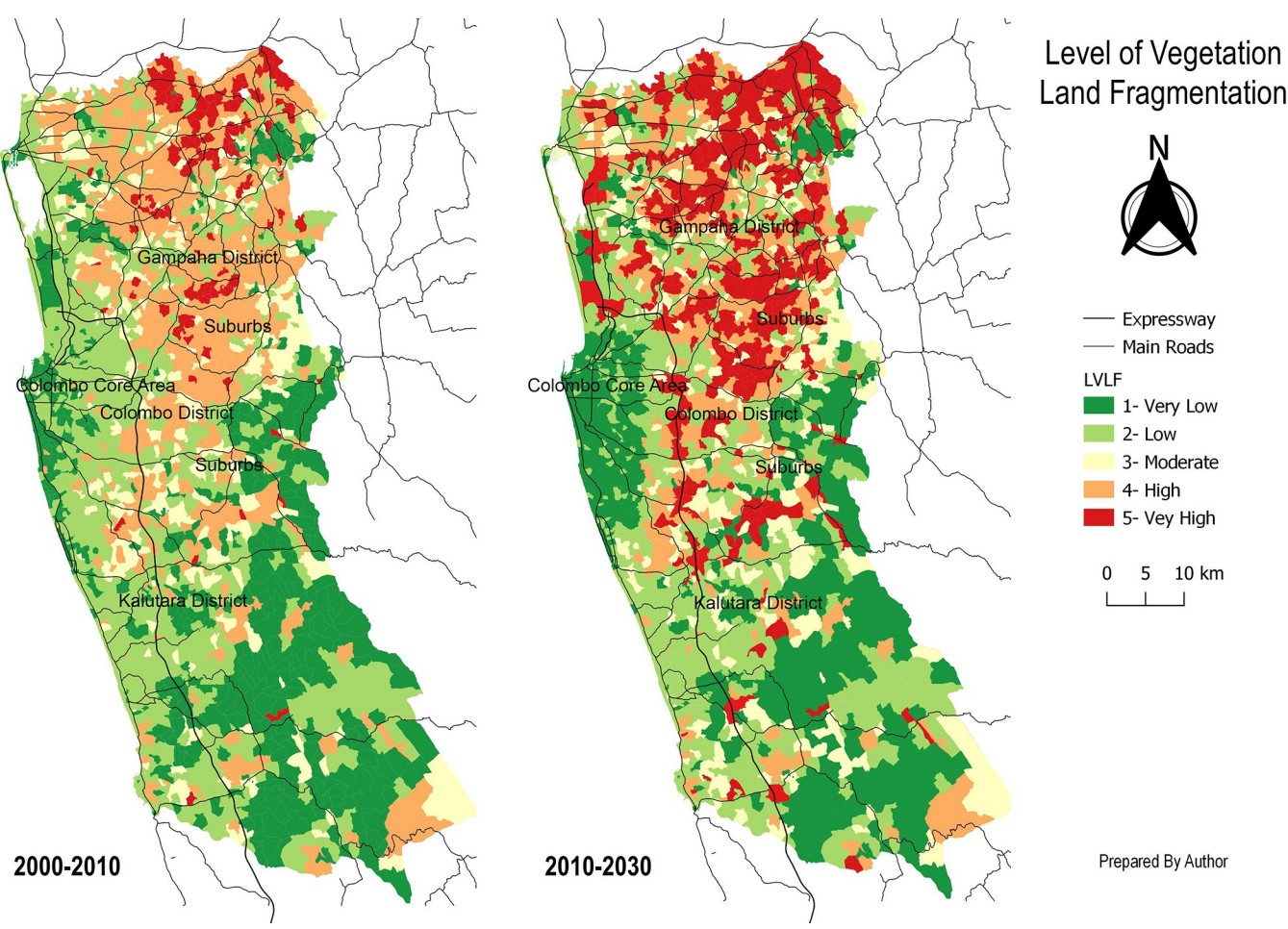

**Fig 14. Map of Future LVLF in Western Province, Sri Lanka.**

findings point to increased accessibility as the primary cause of vegetation fragmentation [36,87]. The findings of this investigation back up the notion of landscape ecology and the fragmentation process as a contribution of this study. With the LVLF, the second-highest positive significance level 23 is associated with vegetation patch association or distance between vegetation patches, which is consistent with the findings of previous research [7,34]. Although earlier research has looked at the configuration factors to quantify the LVLF [3,88,89], this study considers them to be factors of vegetation land fragmentation, as the configuration of the vegetation patches always causes fragmentation. Therefore, the study contributes our understanding of how to account for patch-related factors when modeling the LVLF, which is another contribution to the existing knowledge Patch association, for example, is regarded as one of the drivers of vegetation fragmentation [89], although it is also considered one of the features of vegetation fragmentation in many studies. Further the findings reveal that the association between patches is a significant factor and that reducing the association would raise the LVLF. Patch association is also important in terms of species movements, resource distribution, and ecosystem service, according to basic landscape ecological theory [89,90]. The hypothesis goes on to say that the loss of connection or association will have a detrimental influence on biodiversity, and it encourages nearby land-use types to be converted [29]. As a result, if vegetation patches are closer together, the likelihood of switching vegetation land use

is lower than if patches are separated by greater distances [29,90]. However, other factors, such as the size of the vegetation patch, might influence the distance between patches [28,29]. As a result, more research is needed to identify the relationship between patch association with fragmentation and the composition features of the patch. On the other hand, the highest level of negatively affecting factors on the LVLF is the size of the vegetation large patch. The findings are consistent with previous research [40]. Other factors that are positively influencing LVLF based on their non-linearity, such as vegetation shape irregularity, vegetation shape circularity (circle or linear), built-up land fragmentation, and vegetation land fragmentation of the initial year, are similar to fundamental landscape ecology explanations [29,30,32]. The building density of the initial year and LVLF, on the other hand, are not in the same direction as prior research explored, which is consistent with the previous study [50]. The influence of urban sprawl in suburban regions might be the cause of this observation [36,91,92]. In comparison to Colombo core regions, suburban areas in Western Province do not have a higher level of building density [93]. However, because of expressway construction and road network expansions in the Western Province, suburban regions have become more accessible. Therefore, new buildings with lower density are emerging [93]. Owing to new developments, existing vegetation patches in suburbs have been greatly fragmented compared to highly densified areas, according to a previous study [93]. Consequently, the study finds that building density and LVLF have a negative relationship. However, carrying out a local level research for future studies to validate these results can yield a more accurate outcome. Significantly, the research shows that corridor-related parameters are more important than patch and matrix, which opens-up further research opportunities in the field of land fragmentation. Both patch and matrix-related parameters, in contrast, have a considerable impact on LVLF. Specifically, rather than measuring several indices to assess fragmentation, this study identifies vegetation fragmentation and LVLF aids in determining the factors and their non-linear interactions. The analysis solely grouped the components under patch-corridor-matrix with numerical values into GND borders in this case.

However, the study does not give exact variables in the form of geographical corridors, patches, or matrixes. Due to this constraint, the model findings for each category cannot be spatially interpreted in this study. Therefore, future research should look into employing exact geographical entities rather than confining itself to the border, as this study did; as it can help to develop a spatial model like Futures, SLEUTH, unlike statistical models. In model applications, the study claims that validation of the ANN displays the strongest level of prediction accuracy, since MSE and RMSE are 0.025 percent and 1.574 percent, respectively. After all, the DT model also shows 12% of RAE and 29% of RRSE. The findings further confirm that similar to environmental studies, the AI model can be used to simulate vegetation land fragmentation [1,2,47]. Furthermore, the future simulation depicts an increasing trend of LVLF along expressways in Sri Lanka's Western Province, demonstrating the relevance of corridor-based variables in promoting LVLF.

The Sri Lankan government has recently placed a greater emphasis on expressway construction, intending to increase regional transportation through huge infrastructure projects [94,95]. In the Western Province, for example, the Southern Expressway, the outer-circular highway, and the Katunayake Expressway have all been completed. The Kandy Expressway is also under development and has begun in the Western Province. Therefore, Western Province serves as the country's transportation center. Consequently, rapid land clearances and divisions associated with expressway constructions may be the primary cause of vegetation land fragmentation in Sri Lanka's Western Province and these findings consist with the study of the effect of social and environmental factors on expressway construction in Sri Lanka [95]. However, the research is confined to the Western Province of Sri Lanka, and it solely looks at

vegetation land fragmentation. Thus, it is critical to investigate a variety of case studies and land-use categories such as forest and paddy separately [28,29] while examining fragmentation research, because the fragmentation level may be determined by a variety of factors depending on the land use type [28,29,95]. It is also important to evaluate the case study's various aspects. Furthermore, by referring to local approaches, modeling findings must be validated in the local context. Therefore, future researchers may use the same technique in dissimilar case study areas at the regional, local, and site levels to generalize the study's findings while focusing on land fragmentation in other land use categories other than vegetation. Additionally, future research can investigate the non-linear relationship between vegetation land fragmentation and AI models other than ANN and DT, as well as other factors of vegetation land fragmentation in different categories, in order to identify new determinants of land fragmentation. It is far more necessary to adhere to geographical rather than administrative boundaries because it has the potential to generate several inaccuracies in the analysis findings. Due to the lack of secondary data, this study relied on administrative borders. However, for extremely accurate findings and to construct a spatial model, future researchers need to incorporate spatial data such as patches, corridors, and matrix.

The main contribution of this study is the development of an artificial intelligence-based simulation framework for simulating vegetation land fragmentation in urban regions with acceptable accuracy. The study provides numerical evidence for a nonlinear relationship between factors and land fragmentation caused by vegetation. The study also develops a tree structure to fully describe the phenomena of vegetation land fragmentation. This study contributes methodologically by utilizing AI-based technologies such as ANN and DT to understand complex, non-linear interactions and assess and simulate vegetation land fragmentation.

## 5. Conclusion

The study was successful in building an AI-based simulation framework for modeling vegetation land fragmentation in urban environments, as well as in attempting to discover variables that influence vegetation land fragmentation through non-linear connections. Supervised feedforward ANN (Deep Learning) was used to identify the drivers of vegetation land fragmentation, understand how they behave, and model future vegetation land fragmentation. The DT (supervised classification) model was used to identify various possible scenarios in vegetation land fragmentation.

The implications of study's findings are valuable for urban planners who want to learn more about vegetation land fragmentation. Researchers interested in vegetation land fragmentation could utilize this work to consider vegetation land fragmentation from an urban planning viewpoint and model it in a new context. In urban and regional planning, forecasting future land-use changes, identifying urban growth patterns, and assessing the influence on natural vegetation are difficult challenges. As a result, planners may successfully simulate those occurrences using the developed modeling framework; and use the developed DT diagrams and rules to get quantitative insight into vegetation land fragmentation phenomena and regulate them with strategic actions. The study discovered that increasing accessibility has a substantial influence on land fragmentation in vegetation.

Furthermore, fragmentation threatens tiny irregular and scattered vegetation patches. Therefore, the planner can devise measures to reduce vegetation land fragmentation, such as creating interconnected vegetation corridors and reducing accessibility expansion in more sensitive vegetation. Therefore, by examining the context's land uses, planners may propose solutions to prevent the impacts of vegetation land fragmentation. It is difficult to assess the efficiency of spatial planning initiatives in terms of landscape change and fragmentation in

contemporary practice. Therefore, the study's findings and method can also be used to evaluate the impacts of existing planning attempts.

Data unavailability and data distortions are major drawbacks of the study. Further, the outcomes of the study cannot be interpreted spatially due to data limitations. Future research can build on this framework in other case study sites with various land use categories. It would also be interesting if future academics could put this system through its paces with diverse AI models. This project will continue the ground verification of the results and the modeling framework. Developing a spatial model, moreover, is vital for successfully interpreting spatial dynamics. Despite limitations, the findings of the study provide a framework for quantifying, analyzing, and modeling vegetation land fragmentation, allowing planners to assess the current situation, forecast future trends, and develop effective spatial planning strategies and land monitoring mechanisms to achieve long-term sustainable development.

## Supporting information

**S1 Data.**
(RAR)

## Author Contributions

**Conceptualization:** Amila Jayasinghe, Niroshan Bandara, Chathura De Silva.

**Data curation:** Nesha Ranaweera.

**Formal analysis:** Nesha Ranaweera.

**Funding acquisition:** Amila Jayasinghe.

**Investigation:** Nesha Ranaweera.

**Methodology:** Amila Jayasinghe, Chethika Abenayake, Chathura De Silva.

**Project administration:** Amila Jayasinghe.

**Resources:** Amila Jayasinghe.

**Software:** Niroshan Bandara.

**Supervision:** Amila Jayasinghe.

**Validation:** Amila Jayasinghe.

**Visualization:** Niroshan Bandara.

**Writing – original draft:** Nesha Ranaweera.

**Writing – review & editing:** Chethika Abenayake.

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
