## [Decision Letter · Decision Letter 0]

22 Feb 2022

PONE-D-21-37017Model vegetation land fragmentation in urban areas: an artificial intelligence-based simulation frameworkPLOS ONE

Dear Dr. Jayasinghe,

Thank you for submitting your manuscript to PLOS ONE. After careful consideration, we feel that it has merit but does not fully meet PLOS ONE’s publication criteria as it currently stands. Therefore, we invite you to submit a revised version of the manuscript that addresses the points raised during the review process.

We look forward to receiving your revised manuscript.

Kind regards,

Ashraf Dewan, PhD

Academic Editor

PLOS ONE

Journal Requirements:

“NO - Include this sentence at the end of your statement: The funders had no role in study design, data collection and analysis, decision to publish, or preparation of the manuscript.”

“The authors would like to acknowledge the Senate Research Committee (SRC) Grant, University of Moratuwa, Sri Lanka (No SRC/ST/2021/XXX).”

 “NO - Include this sentence at the end of your statement: The funders had no role in study design, data collection and analysis, decision to publish, or preparation of the manuscript.”

7. We note that Figure 1, 2, 3 and 15 in your submission contain map images which may be copyrighted. All PLOS content is published under the Creative Commons Attribution License (CC BY 4.0), which means that the manuscript, images, and Supporting Information files will be freely available online, and any third party is permitted to access, download, copy, distribute, and use these materials in any way, even commercially, with proper attribution. For these reasons, we cannot publish previously copyrighted maps or satellite images created using proprietary data, such as Google software (Google Maps, Street View, and Earth). For more information, see our copyright guidelines: http://journals.plos.org/plosone/s/licenses-and-copyright.

 a. You may seek permission from the original copyright holder of Figure 1, 2, 3 and 15 to publish the content specifically under the CC BY 4.0 license. 

8. We note you have included a table to which you do not refer in the text of your manuscript. Please ensure that you refer to Table 2 in your text; if accepted, production will need this reference to link the reader to the Table.

Additional Editor Comments:

I have now received comments on your submission. Based on two reviewers’ comments, I now invite you revise your work.

Reviewers' comments:

Reviewer's Responses to Questions

**Comments to the Author**

1. Is the manuscript technically sound, and do the data support the conclusions?

Reviewer #1: Partly

Reviewer #2: Yes

2. Has the statistical analysis been performed appropriately and rigorously? 

Reviewer #1: I Don't Know

Reviewer #2: Yes

3. Have the authors made all data underlying the findings in their manuscript fully available?

Reviewer #1: Yes

Reviewer #2: Yes

4. Is the manuscript presented in an intelligible fashion and written in standard English?

Reviewer #1: No

Reviewer #2: No

5. Review Comments to the Author

Reviewer #1: The authors should attend to the following

Abstract

-There is a difference between vegetation fragmentation and land (landscape) fragmentation? Authors should stick to standard terminology of either vegetation fragmentation or land/landscape fragmentation. If they are interested in understand fragmentation of various land cover or land uses (built-up, wetlands, forests etc) then they should stick to landscape or land fragmentation. If they are interested in the fragmentation of grasslands, forests and other green spaces they should stick to vegetation fragmentation

-Why the authors jumped to mention that,’’ By addressing current research gaps, the objective of this study is to develop an AI-based simulation framework to simulate vegetation land fragmentation in urban areas’’ without specifically highlighting the specific research gap?

-The authors should mention R as a statistical software ie R (statistical software) or R statistical software

Introduction

Please use the standard terminology and avoid ‘‘vegetation land-use changes’’. There is nothing called vegetation land-use change/s

The use of numbering in citation should be consistent. In the number reference system, a number is normally added in parentheses or square brackets in the appropriate place in the text, starting the numbering from 1. Furthermore, the reference bibliography section of the research paper or manuscript is arranged by the order in which the citations appear in the text. Please check again and stick to journal guidelines.

Line 52-61, vegetation fragmentation or land fragmentation is poorly defined. It be should be brief, succinct and not long and winding especially in a journal article

Line 74-78, Again, this statement ‘‘Vegetation land fragmentation refers to variations in shape, size, composition, and distribution of vegetation. As a result, researchers use landscape metrics such as the division index, patch density, number of patches, area-weighted index, and others to evaluate vegetation land fragmentation [7][21]. As a result, it's critical to define vegetation land fragmentation before attempting to quantify..’’ is a repetition of the definition of vegetation or land fragmentation (ie Line 52-61).

Line 111-119, the statement, ‘‘therefore, based on landscape ecology… The Patch-Corridor-Matrix model, which is the essential approach for quantifying vegetation land fragmentation in landscape metrics, is depicted in Fig 2’’ is a repetition of the definition of vegetation or land fragmentation

Line 121, Use Standard English grammar. It should be non- fragmented. Delete not fragmented

Why is it necessary to model and simulate vegetation or land fragmentation? What’s the societal and scientific relevance and contribution? Is not highlighted in the introduction section

Line 146- delete overall methodology used standard terminology

Line 155 -156, ‘‘before defining the models, Pearson correlation was used to exclude the multi-correlated and least-correlated factor’’ is not correct. Note that Pearson correlation a measure of the direction of relationship that exists between two continuous variables. I expected the authors to mention Correlation matrix or Principal Components Analysis in excluding the multi-correlated and least-correlated factor’ ’Correct this

Line 161- Fig 4.should indicate that it’s a flowchart showing the steps or research methodology used in the study. Please revisit Fig 4 caption and better delete it ‘‘the overall method of the study‘‘is not catchy

The research methodology is too repetitive, confusing, long and winding and poorly written.

The discussion part of the manuscript is poorly written. It does not show depth. The discussion section is where you explore the underlying meaning of your research (ie modelling and simulating vegetation fragmentation or land fragmentation) by citing various sources, its possible implications in other areas of study, and the possible improvements that can be made in order to further develop the concerns of your research. How do you compare your results with those from other studies: Are they consistent? If not, discuss possible reasons for the difference. Discuss how could your findings be applied and extend the findings of previous studies?

Reviewer #2: As I read the manuscript carefully, I found some areas in which I would have appreciated greater clarity. I believe the paper could be further strengthened by changes which should be made before it gets published as follows:

The title

As the title of any paper should attract the reader and concisely reflects the actual content of the researchers’ contribution, I found the title of this manuscript needs some modification. For example, I suggest the following title “Modelling vegetation land fragmentation in urban areas of Western Province, Sri Lanka using an artificial intelligence-based simulation technique”

The Abstract

While the abstract summaries the main idea of fragmentation as a landscape ecology term properly, several keys are missing from it mainly the research aims. Please directly state the main aim and objectives (or the research questions) of the study and write about the policy implications of the study area. Likewise, I suggest inserting a couple of sentences to mention all variables that have been incorporated in the modelling process.

Introduction

Line 44 what does the number 392249 refer to and it seems that the citation is written in an integrated or standard way?

Line 62-63 the sentence needs to rewritten.

Line 64 the authors have mentioned “studies point to….” while they have cited only one study (32).

Overall, the introduction section involves both a theoretical framework and related work as well as introducing the main idea of the research. Nevertheless, it was NOT written in cohesion and logical flow style. Similarly, the analytical procedures, utilized spatial techniques for the analysis of land use land cover dynamics have not been made clear.

Theoretical Framework

Besides the poor scientific literature about fragmentation of lands, spatial patterns and ecological process, the literature review section also lacks an adequate description of methods and analyses that previously used in studying several land use land cover LULC applications particularly outside the study area and region. I suggest, to expand the literature review by including additional references particularly the following:

- Spatial disparity patterns of green spaces and buildings in arid urban areas

- Cultivated Land Fragmentation and Its Influencing Factors Detection: A Case Study in Huaihe River Basin, China

- The evolution of urban sprawl: Evidence of spatial heterogeneity and increasing land fragmentation

Other issues in this section should be resolved as follows:

o The authors should combine the two sections “the introduction and theoretical framework” into one section entitled “introduction and literature review”.

o The authors should explain to the reader what motivated this research and why it is important to assess the vegetation land fragmentation in this area specifically.

o Similarly, and from a methodological perspective, various questions should be answered such as how the previous research analyzed and investigated vegetation land fragmentation in Sir Lanka and outside? What are the main spatial methods and techniques that have been used?

o The literature review is very poor and must be extended to include several studies that cover various issues that are relevant to this research.

Methods

The title “method” should be changed into “material and methods”.

I am wondering why the authors have not represented the identified parameters in a map. The authors should provide the readers with a map illustrates all spatial variables utilized in the analysis processes.

Line 187 the explanation of the equation’s symbols should be placed after NOT before.

Findings

The authors should represent the spatial outputs of the modelling process. For instance, the reader expects to see one map for each category such as corridor, patch, and matrix.

The authors should add at least two maps for more clarity of the finding’s representation.

Figure 12. “Rules of variables according to lekprofile algorithm” should be reproduced in a higher level of resolution.

Overall, the results section lacks any sort of cause-and-effect relationships investigation. Consequently, the results have been written in methodological way and thus the authors must modify this section based on spatial patterns distribution across the study area.

Discussion and conclusions

The discussion should tell the story of the science and answer the “why” question of the results.

The explanation is not adequately supported by pieces of evidence, reasoning, and thus, after carrying out the suggested analyses, a major revision is required for this section to clarify the discussion part and link it to the previous research.

The authors should mention the study’s limitations and drawbacks.

Many grammatical, stylistic and syntax errors are found across the manuscript.

6. PLOS authors have the option to publish the peer review history of their article (what does this mean?). If published, this will include your full peer review and any attached files.

Reviewer #1: **Yes: **Pedzisai Kowe

Reviewer #2: No

---

## [Author Response · Author response to Decision Letter 0]

10 Jun 2022

Dear Editor, 

Figures 5 and 15 developed based on the data extracted from http://www.riskinfo.lk/layers/?limit=10&offset=0 and https://www.nsdi.gov.lk/

Hope this compatible with your copyright license 

Thanks and regards

Amila

Dear Editor, 

Thank you very much for your comments. I'm sending this response in response to the comment that we received on 30th May 2022, which is mentioned below.

Regarding these figures, we again kindly ask that you clarify the following points:

a) Where did the authors obtain the maps in Figures 5 and 15?

We'd like to point out that Figures 5 and 15 are the results of our study. They are the property of the paper's authors developed based on analysis.

I'd appreciate it if you could give us your perspective on how to resolve the above-mentioned problem. 

We are looking forward to hearing from you. 

Thank you very much

Best wishes,

Amila Jayasinghe

Dear Editor, 

Thank you so much for your further remarks. All three inaccuracies in the email have been addressed.

1. The cover letter has been revised.

2. Affiliation has been added to the manuscript.

3. Figure 3 has been deleted.

1. "NO authors have competing interests"

 This information should be included in your cover letter or in the "Author Comments" box; we will change the online submission form on your behalf.

2. Please ensure that you include a title page within your main document. You should list all authors and all affiliations as per our author instructions and clearly indicate the corresponding author.

3. We note your response to the copyright query: "We have removed Figures 1, 2, 3 and 15 and supplied replacement figures."

Thank you for providing replacement figures, however, please clarify the following about Figure 3.

Thanks for all comments.

Best wishes,

Amila Jayasinghe

Response to Reviewers 2nd round

Dear Editor, 

Thank you so much for your further remarks. All three inaccuracies in the email have been addressed.

1. The cover letter has been revised.

2. Affiliation has been added to the manuscript.

3. Figure 3 has been deleted.

1. "NO authors have competing interests"

 This information should be included in your cover letter or in the "Author Comments" box; we will change the online submission form on your behalf.

2. Please ensure that you include a title page within your main document. You should list all authors and all affiliations as per our author instructions and clearly indicate the corresponding author.

3. We note your response to the copyright query: "We have removed Figures 1, 2, 3 and 15 and supplied replacement figures."

Thank you for providing replacement figures, however, please clarify the following about Figure 3.

Thanks for all comments.

Best wishes,

Amila Jayasinghe

Response to Reviewers 1st round

Dear Editor,

We would like to thank the Journal of PLOS ONE for giving us the opportunity to revise Manuscript ID: PONE-D-21-37017. We thank the reviewers for their constructive comments. We have carefully taken their comments into consideration in preparing our revision. Below is our response to their comments.

Thanks for all comments.

Best wishes,

Amila Jayasinghe

Response to Editor/Journal Requirements

1.Please ensure that your manuscript meets PLOS ONE's style requirements, including those for file naming.

Response: We have rearranged the manuscript according to the PLOS ONE's style requirements.

Response: We have employed a professional scientific editing service. The country’s current economic crisis and restrictions on international money transfer prevent authors to obtain international services for language editing. However, if reviewers feel further the importance of editing language, we would like to do it before the final submission. 

Name of the colleague: Shereen Rodrigo

Email: srodriggo@gmail.com

Response: We have included it in the submission form. 

“NO - Include this sentence at the end of your statement: The funders had no role in study design, data collection and analysis, decision to publish, or preparation of the manuscript.”

Response: We have included it in the submission form.

“The authors would like to acknowledge the Senate Research Committee (SRC) Grant, University of Moratuwa, Sri Lanka (No SRC/ST/2021/XXX).”

Response: We have removed it from the manuscript. 

 “NO - Include this sentence at the end of your statement: The funders had no role in study design, data collection and analysis, decision to publish, or preparation of the manuscript.”

Response: We have included it in the submission form.

6. In your Data Availability statement, you have not specified where the minimal data set underlying the results described in your manuscript can be found. PLOS defines a study's minimal data set as the underlying data used to reach the conclusions drawn in the manuscript and any additional data required to replicate the reported study findings in their entirety. All PLOS journals require that the minimal data set be made fully available. 

Upon re-submitting your revised manuscript, please upload your study’s minimal underlying data set as either Supporting Information files or to a stable, public repository and include the relevant URLs, DOIs, or accession numbers within your revised cover letter.

Response: We have shared the link of the study’s minimal underlying data set. Data (file://DESKTOP-EF7O048/Data)

7. We note that Figure 1, 2, 3 and 15 in your submission contain map images which may be copyrighted. All PLOS content is published under the Creative Commons Attribution License (CC BY 4.0), which means that the manuscript, images, and Supporting Information files will be freely available online, and any third party is permitted to access, download, copy, distribute, and use these materials in any way, even commercially, with proper attribution. For these reasons, we cannot publish previously copyrighted maps or satellite images created using proprietary data, such as Google software (Google Maps, Street View, and Earth).

We require you to either (1) present written permission from the copyright holder to publish these figures specifically under the CC BY 4.0 license, or (2) remove the figures from your submission. 

Response: We have removed Figures 1, 2, 3 and 15 and supplied replacement figures. 

8. We note you have included a table to which you do not refer in the text of your manuscript. Please ensure that you refer to Table 2 in your text; if accepted, production will need this reference to link the reader to the Table.

Response: We have changed it to Table 2. 

Reviewers' comments:

1. Is the manuscript technically sound, and do the data support the conclusions?

Response: We have improved the manuscript as per the reviewers' comments. 

2. Has the statistical analysis been performed appropriately and rigorously?

Response: We have improved the manuscript as per the reviewers' comments.

3. Have the authors made all data underlying the findings in their manuscript fully available?

Response: We have shared the link of the study’s minimal underlying data set. Data (file://DESKTOP-EF7O048/Data)

4. Is the manuscript presented in an intelligible fashion and written in standard English?

Response: We have employed a professional scientific editing service.

Reviewer #1

Abstract

There is a difference between vegetation fragmentation and land (landscape) fragmentation? Authors should stick to standard terminology of either vegetation fragmentation or land/landscape fragmentation. If they are interested in understand fragmentation of various land cover or land uses (built-up, wetlands, forests etc) then they should stick to landscape or land fragmentation. If they are interested in the fragmentation of grasslands, forests and other green spaces they should stick to vegetation fragmentation. 

Response: We have explained vegetation fragmentation in the entire manuscript. 

Why the authors jumped to mention that,’’ By addressing current research gaps, the objective of this study is to develop an AI-based simulation framework to simulate vegetation land fragmentation in urban areas’’ without specifically highlighting the specific research gap?

Response: The highlighting research gap is unavailability of AI-based simulation framework to simulate vegetation land fragmentation. We have rearranged it by removing “By addressing current research gap”. 

The authors should mention R as a statistical software ie R (statistical software) or R statistical software.

Response: We have changed it to statistical software. 

Introduction

Please use the standard terminology and avoid ‘‘vegetation land-use changes’’. There is nothing called vegetation land-use change/s

Response: We have changed it to “vegetation cover changes”. 

The use of numbering in citation should be consistent. In the number reference system, a number is normally added in parentheses or square brackets in the appropriate place in the text, starting the numbering from 1. Furthermore, the reference bibliography section of the research paper or manuscript is arranged by the order in which the citations appear in the text. Please check again and stick to journal guidelines.

Response: We have corrected the errors in citations and references. 

Line 52-61, vegetation fragmentation or land fragmentation is poorly defined. It be should be brief, succinct and not long and winding especially in a journal article. 

Response: We have clearly defined vegetation fragmentation as the division of vegetation patches into smaller ones. 

Line 74-78, Again, this statement ‘‘Vegetation land fragmentation refers to variations in shape, size, composition, and distribution of vegetation. As a result, researchers use landscape metrics such as the division index, patch density, number of patches, area-weighted index, and others to evaluate vegetation land fragmentation [7][21]. As a result, it's critical to define vegetation land fragmentation before attempting to quantify..’’ is a repetition of the definition of vegetation or land fragmentation (ie Line 52-61).

Response: We have removed the repetitions and reordered the introduction section. 

Line 111-119, the statement, ‘‘therefore, based on landscape ecology… The Patch-Corridor-Matrix model, which is the essential approach for quantifying vegetation land fragmentation in landscape metrics, is depicted in Fig 2’’ is a repetition of the definition of vegetation or land fragmentation.

Response: We have removed the repetitions and reordered the introduction section. 

Line 121, Use Standard English grammar. It should be non- fragmented. Delete not fragmented. 

Response: We have corrected it. 

Why is it necessary to model and simulate vegetation or land fragmentation? What’s the societal and scientific relevance and contribution? Is not highlighted in the introduction section. 

Response: We have respecified the need and contribution in the introduction section. 

Line 146- delete overall methodology used standard terminology

Response: We have changed it to ‘research methodology and techniques’. 

Line 155 -156, ‘‘before defining the models, Pearson correlation was used to exclude the multi-correlated and least-correlated factor’’ is not correct. Note that Pearson correlation a measure of the direction of relationship that exists between two continuous variables. I expected the authors to mention Correlation matrix or Principal Components Analysis in excluding the multi-correlated and least-correlated factor’ ’Correct this

Response: We have corrected it. 

Line 161- Fig 4.should indicate that it’s a flowchart showing the steps or research methodology used in the study. Please revisit Fig 4 caption and better delete it ‘‘the overall method of the study‘‘is not catchy. 

Response: We have changed it to ‘Steps of research methodology’. 

The research methodology is too repetitive, confusing, long and winding and poorly written.

Response: We have removed the repetitions and We have employed a professional scientific editing service to enrich the content. 

The discussion part of the manuscript is poorly written. It does not show depth. The discussion section is where you explore the underlying meaning of your research (ie modelling and simulating vegetation fragmentation or land fragmentation) by citing various sources, its possible implications in other areas of study, and the possible improvements that can be made in order to further develop the concerns of your research. How do you compare your results with those from other studies: Are they consistent? If not, discuss possible reasons for the difference. Discuss how could your findings be applied and extend the findings of previous studies?

Response: We have rewritten the discussion section including comparisons between prior studies and explained possible reasons. 

Reviewer #2

The title

As the title of any paper should attract the reader and concisely reflects the actual content of the researchers’ contribution, I found the title of this manuscript needs some modification. For example, I suggest the following title “Modelling vegetation land fragmentation in urban areas of Western Province, Sri Lanka using an artificial intelligence-based simulation technique”

Response: We have changed the title. 

The Abstract

While the abstract summaries the main idea of fragmentation as a landscape ecology term properly, several keys are missing from it mainly the research aims. Please directly state the main aim and objectives (or the research questions) of the study and write about the policy implications of the study area. Likewise, I suggest inserting a couple of sentences to mention all variables that have been incorporated in the modelling process.

Response: We have rewritten the abstract including the aim, objectives, and variables. 

Introduction

Line 44 what does the number 392249 refer to and it seems that the citation is written in an integrated or standard way?

Response: We have corrected the errors in citations and references.

Line 62-63 the sentence needs to rewritten.

Response: We have rewritten it and have employed a professional scientific editing service to improve English. 

Line 64 the authors have mentioned “studies point to….” while they have cited only one study (32).

Response: We have corrected it. 

Overall, the introduction section involves both a theoretical framework and related work as well as introducing the main idea of the research. Nevertheless, it was NOT written in cohesion and logical flow style. Similarly, the analytical procedures, utilized spatial techniques for the analysis of land use land cover dynamics have not been made clear.

Response: We have strengthened the introduction section including spatial techniques. Also, we have restructured the introduction section. 

Theoretical Framework

Besides the poor scientific literature about fragmentation of lands, spatial patterns and ecological process, the literature review section also lacks an adequate description of methods and analyses that previously used in studying several land use land cover LULC applications particularly outside the study area and region. I suggest, to expand the literature review by including additional references particularly the following:

- Spatial disparity patterns of green spaces and buildings in arid urban areas

- Cultivated Land Fragmentation and Its Influencing Factors Detection: A Case Study in Huaihe River Basin, China

- The evolution of urban sprawl: Evidence of spatial heterogeneity and increasing land fragmentation

Response: We have included the above studies and expanded the literature review by including additional references. 

Other issues in this section should be resolved as follows:

o The authors should combine the two sections “the introduction and theoretical framework” into one section entitled “introduction and literature review”.

The authors should explain to the reader what motivated this research and why it is important to assess the vegetation land fragmentation in this area specifically.

o Similarly, and from a methodological perspective, various questions should be answered such as how the previous research analyzed and investigated vegetation land fragmentation in Sir Lanka and outside? What are the main spatial methods and techniques that have been used?

o The literature review is very poor and must be extended to include several studies that cover various issues that are relevant to this research.

Response: We have combined the two sections “the introduction and theoretical framework” into one section entitled “introduction and literature review”.

Methods

The title “method” should be changed into “material and methods”.

Response: We have corrected it. 

I am wondering why the authors have not represented the identified parameters in a map. The authors should provide the readers with a map illustrates all spatial variables utilized in the analysis processes.

Response: We have included an additional Figure showing the spatial distribution of factors. 

Line 187 the explanation of the equation’s symbols should be placed after NOT before.

Response: We have corrected it. 

Findings

The authors should represent the spatial outputs of the modelling process. For instance, the reader expects to see one map for each category such as corridor, patch, and matrix. 

The authors should add at least two maps for more clarity of the finding’s representation.

Response: Due to a lack of secondary data, this study investigated data under GNDs boundaries, which is the local level administrative boundary. This is one of the study's weaknesses. As a result, the results of this study cannot be interpreted in terms of corridor, patch, or matrix. Future research will, however, overcome this constraint. We've made it clear that there's a limitation.

Figure 12. “Rules of variables according to lekprofile algorithm” should be reproduced in a higher level of resolution.

Response: We have reproduced it in a higher level of resolution. 

Overall, the results section lacks any sort of cause-and-effect relationships investigation. Consequently, the results have been written in methodological way and thus the authors must modify this section based on spatial patterns distribution across the study area.

Response: We have improved the figures indicating spatial references. 

Discussion and conclusions

The discussion should tell the story of the science and answer the “why” question of the results.

Response: We have specified the reasons for the results. 

The explanation is not adequately supported by pieces of evidence, reasoning, and thus, after carrying out the suggested analyses, a major revision is required for this section to clarify the discussion part and link it to the previous research.

Response: We have cited multiple studies to provide evidence and reasons for results. Also, we have provided suggestions for future studies. 

The authors should mention the study’s limitations and drawbacks.

Response: We have specified the limitations in the discussion section and the summary in the conclusion section. 

Many grammatical, stylistic and syntax errors are found across the manuscript.

Response: We have corrected grammatical, stylistic and syntax errors.

---

## [Decision Letter · Decision Letter 1]

18 Jul 2022

PONE-D-21-37017R1Modelling vegetation land fragmentation in urban areas of Western Province, Sri Lanka using an artificial intelligence-based simulation techniquePLOS ONE

Dear Dr. Jayasinghe,

Thank you for submitting your manuscript to PLOS ONE. After careful consideration, we feel that it has merit but does not fully meet PLOS ONE’s publication criteria as it currently stands. Therefore, we invite you to submit a revised version of the manuscript that addresses the points raised during the review process.

We look forward to receiving your revised manuscript.

Kind regards,

Ashraf Dewan, PhD

Academic Editor

PLOS ONE

Reviewers' comments:

Reviewer's Responses to Questions

**Comments to the Author**

1. If the authors have adequately addressed your comments raised in a previous round of review and you feel that this manuscript is now acceptable for publication, you may indicate that here to bypass the “Comments to the Author” section, enter your conflict of interest statement in the “Confidential to Editor” section, and submit your "Accept" recommendation.

Reviewer #1: (No Response)

Reviewer #3: (No Response)

2. Is the manuscript technically sound, and do the data support the conclusions?

Reviewer #1: Yes

Reviewer #3: Partly

3. Has the statistical analysis been performed appropriately and rigorously? 

Reviewer #1: Yes

Reviewer #3: Yes

4. Have the authors made all data underlying the findings in their manuscript fully available?

Reviewer #1: Yes

Reviewer #3: Yes

5. Is the manuscript presented in an intelligible fashion and written in standard English?

Reviewer #1: Yes

Reviewer #3: Yes

6. Review Comments to the Author

Reviewer #1: Authors should avoid providing a definition of a term/terms (ie vegetation/fragmentation fragmentation) in the abstract. Instead a strong and brief background of the topic in few lines is most appropriate.

The section ‘’ Research methodology and techniques” should be numbered and combined with section Materials and methods. Its appropriate to use “Methods and Materials” terminology. However the authors are encouraged to stick to the Journal guidelines

All sections (introduction, Materials and methods, Results and Discussion and Conclusion) in the manuscript should be appropriately number

The Conclusion is very long and should be shortened?.

In the Conclusion section it is not merely an issue of summarizing the findings but what is the implication of the research findings and the methods developed in this study?

Reviewer #3: I have been invited to review the revised version (revision 1) and I can see that authors have attempted to improved the original submission. However, still there a few issues in the revision. I would therefore suggest to address the following issues:

[1] Show international readership and significance of the work. In particular, you are to draw examples of other cities to show how degradation of urban vegetation is leading to various issues in urban areas such as deterioration of ecosystems services, flooding, heat island etc. These works may be of help https://doi.org/10.1038/s41467-021-26887-4;
https://doi.org/10.1016/j.ecoinf.2022.101730;
https://doi.org/10.1016/j.ufug.2021.127128;
https://doi.org/10.1088/1748-9326/abef8e;
https://doi.org/10.1016/j.apgeog.2021.102533

[2] Avoid ‘redefining’ fragmented or non-fragmented vegetation throughout as this is unclear in your work

[3] Number sections and subsections. Change ‘Method’ to “Data and methods’. I would remove equations that you placed in the manuscript. You can have them in this work if they are developed by you.

[4] Improve and enhance discussion section to demonstrate how this work contributes to the extant knowledgebase. Why conclusion section is too large? Be brief and provide take home messages.

7. PLOS authors have the option to publish the peer review history of their article (what does this mean?). If published, this will include your full peer review and any attached files.

Reviewer #1: **Yes: **PEDZISAI KOWE

Reviewer #3: No

---

## [Author Response · Author response to Decision Letter 1]

15 Sep 2022

Response to Reviewers

Dear Editor,

We would like to thank the Journal of PLOS ONE for giving us the opportunity to revise Manuscript ID: PONE-D-21-37017R1. We thank the reviewers for their constructive comments. We have carefully taken their comments into consideration in preparing our revision. Below is our response to their comments.

Thanks for all comments.

Best wishes,

Amila Jayasinghe

Reviewers' comments:

Reviewer #1

Authors should avoid providing a definition of a term/terms (ievegetation/fragmentation fragmentation) in the abstract. Instead a strong and brief background of the topic in few lines is most appropriate.

Response: We have avoided the definition and included a general statement. 

The section ‘’ Research methodology and techniques” should be numbered and combined with section Materials and methods. Its appropriate to use “Methods and Materials” terminology. However the authors are encouraged to stick to the Journal guidelines

Response: We have numbered the sections in the method and changed the section name to “Materials and methods” as per the journal guidelines. 

All sections (introduction, Materials and methods, Results and Discussion and Conclusion) in the manuscript should be appropriately number

Response: We have numbered all the sections. 

The Conclusion is very long and should be shortened?.

Response: We have shortened the conclusion section. 

In the Conclusion section it is not merely an issue of summarizing the findings but what is the implication of the research findings and the methods developed in this study?

Response: We have improved the conclusion section by highlighting the implications and methods involved in this study. 

Reviewer #3

[1] Show international readership and significance of the work. In particular, you are to draw examples of other cities to show how degradation of urban vegetation is leading to various issues in urban areas such as deterioration of ecosystems services, flooding, heat island etc. These works may be of help https://doi.org/10.1038/s41467-021-26887-4;
https://doi.org/10.1016/j.ecoinf.2022.101730;
https://doi.org/10.1016/j.ufug.2021.127128;
https://doi.org/10.1088/1748-9326/abef8e;
https://doi.org/10.1016/j.apgeog.2021.102533

Response: We have included the above-mentioned studies in the introduction section as some case studies. 

[2] Avoid ‘redefining’ fragmented or non-fragmented vegetation throughout as this is unclear in your work

Response: We have removed the figure and the explanation. 

[3] Number sections and subsections. Change ‘Method’ to “Data and methods’. I would remove equations that you placed in the manuscript. You can have them in this work if they are developed by you.

Response: We have numbered the method section and we have added the references of the equation to acknowledge them. 

[4] Improve and enhance discussion section to demonstrate how this work contributes to the extant knowledgebase. Why conclusion section is too large? Be brief and provide take home messages.

Response: We have improved the discussion section and we specified the implication of the study in the conclusion.

---

## [Editor Report · Decision Letter 2]

19 Sep 2022

Modelling vegetation land fragmentation in urban areas of Western Province, Sri Lanka using an artificial intelligence-based simulation technique

PONE-D-21-37017R2

Dear Dr. Jayasinghe,

We’re pleased to inform you that your manuscript has been judged scientifically suitable for publication and will be formally accepted for publication once it meets all outstanding technical requirements.

Kind regards,

Ashraf Dewan, PhD

Academic Editor

PLOS ONE
---

## [Editor Report · Acceptance letter]

22 Sep 2022

PONE-D-21-37017R2 

Modelling vegetation land fragmentation in urban areas of Western Province, Sri Lanka using an Artificial Intelligence-based simulation technique 

Dear Dr. Jayasinghe:

I'm pleased to inform you that your manuscript has been deemed suitable for publication in PLOS ONE. Congratulations! Your manuscript is now with our production department. 

Kind regards, 

on behalf of

Dr. Ashraf Dewan 

Academic Editor

PLOS ONE